# Carbon Capture from CO₂-Rich Natural Gas via Gas-Liquid Membrane Contactors with Aqueous-Amine Solvents: A Review

**Guilherme Pereira da Cunha** , **José Luiz de Medeiros** * and **Ofélia de Queiroz F. Araújo**

Escola de Química, Federal University of Rio de Janeiro, CT, E, Ilha do Fundão, Rio de Janeiro 21941-909, RJ, Brazil
* Correspondence: jlm@eq.ufrj.br

**Abstract:** Gas–liquid membrane contactor is a promising process intensification technology for offshore natural gas conditioning in which weight and footprint constraints impose severe limitations. Thanks to its potential for substituting conventional packed/trayed columns for acid-gas absorption and acid-gas solvent regeneration, gas-liquid membrane contactors have been investigated experimentally and theoretically in the past two decades, wherein aqueous-amine solvents and their blends are the most employed solvents for carbon dioxide removal from natural gas in gas-liquid membrane contactors. These efforts are extensively and critically reviewed in the present work. Experimentally, there are a remarkable lack of literature data in the context of gas–liquid membrane contactors regarding the following topics: water mass transfer; outlet stream temperatures; head-loss; and light hydrocarbons (e.g., ethane, propane, and heavier) mass transfer. Theoretically, there is a lack of complete models to predict gas-liquid membrane contactor operation, considering multicomponent mass balances, energy balances, and momentum balances, with an adequate thermodynamic framework for correct reactive vapor–liquid equilibrium calculation and thermodynamic and transport property prediction. Among the few works covering modeling of gas-liquid membrane contactors and implementation in professional process simulators, none of them implemented all the above aspects in a completely successful way.

**Keywords:** gas–liquid membrane contactor; natural gas conditioning; carbon capture; solvent regeneration; process intensification; process simulation

---

## 1. Introduction

The global population is continuously increasing, and it is foreseen that it will exceed 10 billion by 2050 [1]. In this scenario, natural gas (NG) exploration and production attract interest in meeting the growing world energy demand [2]. Moreover, NG plays a key role in the transition to renewable energy sources [3] as NG is the cleanest fossil fuel due to its high hydrogen/carbon ratio [4].

Proven NG reserves around the world are approximately 196.9 trillion cubic meters [4], but onshore NG reserves—with simpler production that is closer to consumers—are decreasing, while offshore NG reserves far from the coast and in deep waters are growing [5]. In this regard, oil production with associated NG in deep-waters and ultra-deep-waters usually presents high carbon dioxide (CO₂) content and gas-to-oil ratio, which entail difficulties for gas processing in offshore rigs where there are weight/footprint limitations [6]. Due to complex logistics for NG production/transportation in the context of deep-water offshore rigs, high costs lead to significant offshore CO₂-rich NG reserves being left unexplored. Albeit the existence of efficient technologies for onshore NG production, innovative low-cost technologies are necessary for deep-waters processing of CO₂-rich NG [5].

The concept of carbon capture utilization and storage (CCUS) is necessary to allow a smooth transition from fossil energy to renewable energy to keep a profitable industrial operation. In CO₂-rich NG processing, CO₂ removed from NG is transported to a storage

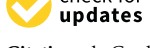

site [7]. Promising $CO_2$ utilizations are enhanced oil recovery (EOR) in oil-and-gas reservoirs [8] or enhanced gas recovery (EGR) in gas reservoirs [9]. In CCUS-EOR or CCUS-EGR, $CO_2$ is injected into reservoirs to increase oil and gas production. If the increase is high enough, the extra revenues can offset $CO_2$ capture/transportation costs entailing profits [7].

### 1.1. Natural Gas Purification Requirements

Raw NG primarily contains methane ($CH_4$) and secondarily ethane, propane, butanes, and heavier hydrocarbons. Other common constituents are $CO_2$, hydrogen sulfide ($H_2S$), water ($H_2O$), nitrogen ($N_2$), and trace components. Raw NG purification is necessary to meet gas pipeline specifications and market/environment regulations [10].

NG processing usually comprises: (i) $H_2S$ removal [11]; (ii) water dew-point adjustment (WDPA) via dehydration [12]; (iii) hydrocarbon dew-point adjustment (HCDPA) [10]; and (iv) $CO_2$ removal. $H_2S$ causes pipeline/equipment corrosion, it is undesirable in combustion, and threatens human health and the environment [11]. WDPA is necessary mainly because water condensed from NG may form solid gas-hydrates under high-pressure and low-temperature typically found in subsea pipelines [12]. HCDPA is necessary to avoid hydrocarbon condensation in pipelines (safety and efficiency issues) and to recover valuable NG liquids [13]. Besides, HCDPA removes hydrocarbons to meet the heating value specification (Wobbe Indexes), ensuring adequate gas–turbine and combustion equipment operations minimizing emissions [5]. Finally, $CO_2$ content is reduced to 2–3 mol%, increasing NG heating value, preventing solid $CO_2$-hydrates, avoiding corrosion, and avoiding occupying pipeline capacity with inert [14].

### 1.2. Carbon Capture from $CO_2$-Rich NG

Several technologies exist for carbon capture from $CO_2$-rich NG. Pressure-swing adsorption (PSA) prescribes selective $CO_2$ adsorption onto solid adsorbents at high-pressure conjugated with low-pressure $CO_2$ desorption [15]. PSA issues comprehend the methane purity-recovery tradeoff [16] and the low-pressure $CO_2$ release entailing compression costs [17].

Cryogenic distillation is considered for $CO_2 \geq 10$ mol% due to lesser energy consumption and footprint compared to chemical absorption [18]. However, since cryogenic distillation operates at temperatures below the $CO_2$ triple-point [19], there is a risk of $CO_2$ freeze-out (dry-ice) inside the equipment, causing blockage [18]. Moreover, high-pressure cryogenic columns are protected against pressure surges by blowdown valves, which under cryogenic conditions can lead to dry-ice formation via the Joule–Thomson effect [20].

In offshore $CO_2$-rich NG processing, membrane permeation (MP) is adopted due to its low footprint, modularity, and flexibility to feed composition changes [21] and is recommended for high $CO_2$ content NG [22]. However, thanks to its capacity-selectivity tradeoff [23], permeate methane losses are remarkable, and the low-pressure $CO_2$-rich permeate imposes high compression costs [24].

Physical absorption (PhA) prescribes a physical solvent that dissolves $CO_2$ at high-pressure [25], requiring high $CO_2$ fugacity for high $CO_2$ loadings [26]. PhA is much less energy-intense for solvent regeneration compared to chemical absorption at the expense of a low-pressure $CO_2$ stripping [6]. PhA drawbacks comprise (i) low $CO_2/CH_4$ selectivity entailing $CH_4$ losses in the stripped gas; (ii) low-pressure stripped gas entailing compression costs for EOR destination.

The most mature technology for $CO_2$ and $H_2S$ removal from $CO_2$-rich NG is chemical absorption (ChA), which is based on $CO_2$ reactions with aqueous-alkanolamine solvents [27]. Aqueous-monoethanolamine (aqueous-MEA) and aqueous-methyl-diethanolamine (aqueous-MDEA) are the most used chemical solvents [28]. The main advantage is the high $CO_2/CH_4$ selectivity in a wide range of operation pressures [27] and this selectivity is particularly high at low $CO_2$ fugacity [22]. ChA generally operates with packing columns that have beds of variously-shaped packings—e.g., Pall rings or Berl saddles—providing a high specific surface area for $CO_2$ transfer. Packing columns are fed with lean solvent at the top

and with raw NG at the bottom in countercurrent flows [29]. ChA drawbacks comprise: (i) high heat-ratio (kJ/kg$^{CO2}$) solvent regeneration [28]; (ii) solvent degradation (oxidation and thermal degradation) causing corrosion and efficiency loss; (iii) low-pressure $CO_2$ stripping entailing compression costs for EOR; (iv) hydraulic issues (e.g., foaming, channeling, flooding and entrainment) [29].

The gas-liquid membrane contactor (GLMC) combines MP and ChA advantages—without the respective drawbacks—becoming a promising solution for $CO_2$-rich NG decarbonation [30]. In GLMC, the hollow-fiber membrane (HFM) provides a non-dispersive gas-liquid contact avoiding packing-columns hydraulic problems like foaming, channeling, flooding, and entrainment [31]. Comparatively, with MP, the selective chemical solvent and thousands of HFM inside the GLMC shell ensure high $CO_2$/$CH_4$ selectivity with low $CH_4$ loss, high mass transfer area per shell, compactness, modularity, and easy scale-up, all desired attributes for $CO_2$ removal from $CO_2$-rich NG with aqueous-amine solvents in offshore rigs [22]. Comparatively, with conventional $CO_2$ absorption packing columns, GLMC reaches reductions up to 70% and 66% in size and weight, respectively [32]. Angular indifference is another advantage of GLMC vis-à-vis the sway of floating offshore rigs [33].

### 1.3. Review Articles on GLMC: Highlights and Gaps

An overview with an appreciation of typical GLMC reviews is conducted. Table 1 condenses reviews on GLMC aspects and developments, while Table 2 condenses reviews on post-combustion carbon capture GLMC, which, despite being somewhat outside the scope—as operating conditions are different from NG processing counterparts—is kept in the analysis due to common particularities of both scenarios (e.g., isothermal modeling issues and low-pressure $CO_2$ stripping GLMC). Table 3 analyzes general reviews on GLMC $CO_2$ removal from NG.

**Table 1.** Review articles: GLMC aspects and developments.

| Description | Reference |
|---|---|
| GLMC environmental applications. | Mansourizadeh et al. [30] |
| HFM and solvent developments for $CO_2$ absorption. | Kim et al. [34] |
| New membranes, solvents, and GLMC configurations. | Li et al. [35] |
| Membrane development. | Xu et al. [36] |
| GLMC applications. | Babin et al. [37] |
| Membrane pore wetting in $CO_2$ capture with GLMC. | Ibrahim et al. [38] |
| GLMC acid-gas removal. | Zhang and Wang [39] |

**Table 2.** Review articles: GLMC post-combustion carbon capture.

| Description | Reference |
|---|---|
| Comparisons of GLMC models. | Rivero et al. [40] |
| GLMC operation and models; highlighted the need for non-isothermal GLMC modeling with water interfacial transfer. | Zhao et al. [41] |
| $CO_2$ stripping GLMC modeling, including state-of-the-art GLMC; solvent selection based on experiments. | Vadillo et al. [42] |
| GLMC modeling (no critical analysis). | Zhang et al. [43] |
| GLMC modeling (no critical analysis). | Cui and deMontigny [44] |
| Isothermal GLMC modeling with chemical reaction; cited eight articles with isothermal GLMC modeling. | Li and Chen [45] |

**Table 3.** Review articles: GLMC $CO_2$ removal from $CO_2$-rich NG.

| Description | Reference |
|---|---|
| Isothermal GLMC modeling without energy balances and assuming ideal gas behavior; poorly discussed head-loss calculation. | Ng et al. [46] |
| Comparison of technologies for offshore rigs: ChA, GLMC, rotating packed-bed, ultrasonic reactor, microchannel reactor. | Tay et al. [29] |
| Comparison of state-of-the-art MP and GLMC. | Siagian et al. [24] |
| MP and GLMC for acid-gas removal and NG dehydration in offshore rigs. | Dalane et al. [47] |
| GLMC applications, including $H_2S$ removal from NG. | Bazhenov et al. [48] |
| Comprehensive review on solvent selection for GLMC NG decarbonation. | Shokrollahi et al. [25] |
| Ionic-liquid solvents for $CO_2/H_2S$ removal from NG; experimental and design studies to make ionic-liquid solvents as effective as aqueous-amine solvents. | Karadas et al. [49] |
| Simplified isothermal GLMC modeling: ideal gas behavior, low pressure, and Henry's law for $CO_2$ vapor-liquid equilibrium (VLE). Out-of-date. | Mansourizadeh and Ismail [50] |

*1.4. The Present Work*

This work intends to fill the gaps in common reviews on GLMC for NG decarbonation and solvent regeneration seen in Table 3. Common deficiencies of Table 3 reviews comprehend: (i) lack of rigorous high-pressure non-isothermal GLMC modeling; (ii) lack of rigorous estimation of thermodynamic properties under multi-reactive conditions; (iii) lack of consideration of thermal and compressibility effects under chemical reactions; and (iv) GLMC implementation in professional process simulators.

In this context, this work critically reviews experimental and theoretical GLMC studies focusing on two applications: (i) $CO_2$ removal from $CO_2$-rich NG considering high-pressure operation and $H_2O/H_2S$ handling and (ii) low-pressure $CO_2$ stripping for solvent regeneration. Section 2 reviews studies on GLMC principles, materials and configurations. Section 3 addresses experimental studies on the two mentioned applications. Section 4 critically reviews GLMC modeling for the two mentioned applications, including GLMC implementation in professional process simulators, analyzing model simplifications and limitations. Future prospects and concluding remarks are the object of Section 5.

## 2. GLMC: Principles, Materials, and Configurations

GLMC is modular equipment configured in batteries. Each GLMC module contains thousands of hydrophobic HFM tubes fixed into a shell-and-tube arrangement [51] in Figure 1. Equipment inclination translates into GLMC angular indifference [33].

GLMC is configurable as an absorber (for $CO_2$ capture) or as a stripper ($CO_2$ stripping for solvent regeneration). In both cases, GLMC has two feeds (where $L$ is the molar flowrate of the GLMC liquid stream and $V$ is the gas counterpart): (i) GLMC-absorber: NG feed $\left(V_{absorber}^{in}\right)$; lean aqueous-amine solvent $\left(L_{absorber}^{in}\right)$ [52]; (ii) GLMC-stripper: Sweep-gas (nitrogen [53] or steam [54]) $\left(V_{stripper}^{in}\right)$; rich aqueous-amine solvent $\left(L_{stripper}^{in}\right)$.

GLMC modules can operate with parallel $V$ and $L$ flows (co-current contact) [55] or opposed $V$ and $L$ flows (counter-current contact) [56]. Whichever the case, $V$ flows inside HFM tubes or on the shell-side, while $L$ respectively flows on the shell-side or inside HFM tubes. Consequently, there are four different GLMC configurations shown in Figure 1: (a,b) apply counter-current contact; (c,d) apply parallel contact; (a,c) have $L$ in shell-side and $V$ inside HFM tubes; (b,d) have $L$ inside HFM tubes and $V$ in shell-side [57]. Since HFM is a physical barrier against phase dispersion, additional phase separation equipment is not necessary [37]. HFM also increases the interfacial area reducing equipment size [58].

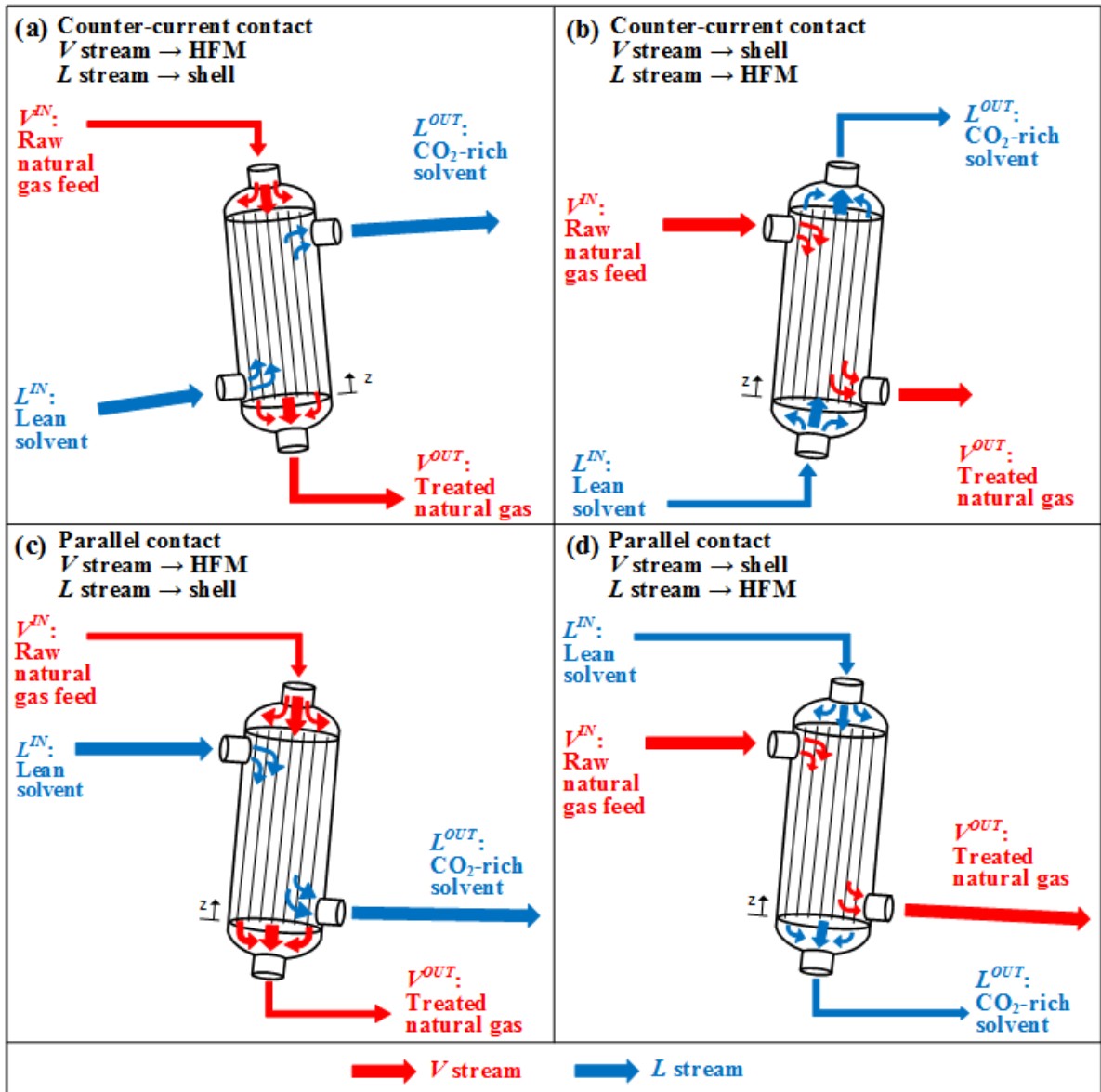

**Figure 1.** GLMC setups: Counter–current contact: (**a**) HFM:*V*, Shell:*L*; (**b**) HFM:*L*, Shell:*V*. Parallel contact: (**c**) HFM:*V*, Shell:*L*; (**d**) HFM:*L*, Shell:*V*.

GLMC is based on reactive-vapor-liquid-equilibrium (RVLE) since *V* and *L* streams are kept in direct contact through the HFM pores and relatively rapid chemical reactions occur in the liquid. In GLMC-absorber with chemical solvent, high $CO_2/CH_4$ selectivity is imposed by the aqueous solvent thanks to $CH_4$ hydrophobicity [33] and selective $CO_2$ conversion via liquid phase chemical reactions. When crossing the membranes, $CH_4$ rapidly reaches its low solubility in the aqueous solvent, such that $CH_4$ bubbles as a small vapor phase may form on the liquid side. As absorption chemical reactions between the solvent and $CO_2$ (or $H_2S$) are exothermic, the liquid temperature continuously increases, reducing $CH_4$ solubility and raising $CH_4$ fugacity over the liquid. Therefore, an *L* stream can be a single-phase liquid or a two-phase stream. Moreover, the $CH_4$ bubbles on the solvent side contribute to stripping some $CO_2/H_2S$ from the solvent, at the same time raising trans-membrane transport via reduction of $CO_2/H_2S$ fugacities in the liquid [33].

On the other hand, in GLMC-absorber with physical solvent, $CO_2$ does not react with the solvent, so species mass transfers and $CO_2/CH_4$ selectivity depend only on gas

diffusivities into the solvent and solute solubilities in the solvent. In this case, $CO_2$ Henry's Law is frequently used for VLE predictions [59].

In GLMC, convective heat transfers occur due to temperature differences between (i) *V* and *L* streams across HFM; (ii) shell-side phase and shell outside. As molar isobaric heat capacity ($\overline{C}_P$) values imply $V\overline{C}_{P_V} << L\overline{C}_{P_L}$, the *L* stream controls the temperature distribution; consequently, *L* and *V* temperature difference is near zero $|T_V - T_L| \approx 0$ at a given GLMC axial position due to convective interfacial heat transfer.

Water can transfer $L\rightarrow V$ or $V\rightarrow L$, depending on the interfacial difference of $H_2O$ fugacities $\hat{f}_{V_{H_2O}} - \hat{f}_{L_{H_2O}}$, influenced by the water content of feeds and absorption heat effects. For counter-current GLMC, the lowest $T_V$, $T_L$ values occur at the *L* inlet, which is also the *V* outlet position. As *V* approaches its outlet at water saturation, its temperature lowers and condensation can occur, turning *V* into a two-phase flow [33]. Additionally, the large absorption heat release can vaporize solvent on the *L* side [60]. Considering all these thermal effects, both *L* and *V* can be single-phase or two-phase streams at some GLMC point [33].

GLMC-stripper regenerates physical solvents (*L*) with nitrogen or steam as sweep-gas (*V*). A vacuum-driven pump can also be employed to reduce V pressure. Since $CO_2/H_2S$ fugacities in the sweep-gas are lower than in the solvent, $CO_2/H_2S$ move $L\rightarrow V$ enabling separation without heat supply [54]. As Figure 2 shows, $CO_2/H_2S$ move $V\rightarrow L$ in GLMC-absorber and $L\rightarrow V$ in GLMC-stripper, while $H_2O$ moves $V\rightarrow L$ or $L\rightarrow V$ whether $\hat{f}_{V_{H_2O}} - \hat{f}_{L_{H_2O}} > 0$ or $\hat{f}_{V_{H_2O}} - \hat{f}_{L_{H_2O}} < 0$. $N_2$ transfer in GLMC-stripper is negligible since $N_2$ is highly hydrophobic [53].

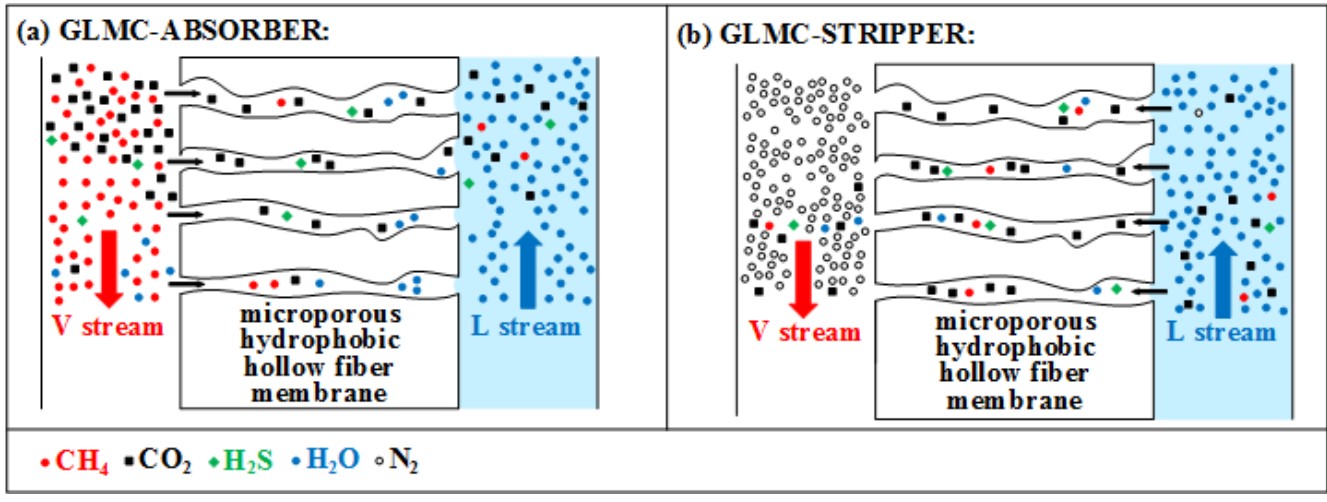

**Figure 2.** Multicomponent mass transfers: GLMC-absorber (**a**); GLMC-stripper (**b**) ($H_2O$ can move $V\rightarrow L$ or $L\rightarrow V$).

### 2.1. HFM for GLMC

Contrarily to MP, which uses dense selective membranes, GLMC adopts non-selective microporous HFM to allow *V-L* contact and selectivity is imposed by the solvent [61]. Figure 3 depicts how HFM works as a mass transfer barrier preventing phase dispersion [62]. Figure 3a represents direct *V-L* contact in tray columns or packing columns with phase dispersion. Figure 3b shows microporous HFM with gas-filled pores ensuring *V-L* contact with low mass transfer resistance [63]. However, during GLMC operation, the solvent may fill the pores (Figure 3c), increasing mass transfer resistance drastically and reducing transfer fluxes. This undesirable phenomenon compromises GLMC performance and is called membrane pore wetting (MPW) [62]. MPW is influenced by membrane characteristics (pores diameter, tortuosity, porosity, surface roughness), solvent surface tension, and operating conditions, while some solvents may favor MPW by affecting HFM

hydrophobicity, morphology, and roughness. To avoid MPW, HFM material must be hydrophobic such as polypropylene (PP), polyvinylidene fluoride (PVDF), polytetrafluorethylene (PTFE), polyether ether ketone (PEEK), and solvents should have high surface tension, negligible vapor pressure and should not chemically attack HFM [41]. Figure 3d shows another solution against MPW wherein dense HFM $CO_2$ permeable and solvent impermeable are used. However, dense HFM presents high transfer resistance and does not fit in the original GLMC concept of non-dispersive *V-L* contact [62]. A compromise solution uses composite HFM (Figure 3e) consisting of an ultrathin dense skin layer onto microporous support. The ultrathin layer is generally a fluorine-based hydrophobic material, while the microporous support—made of a cheaper material like PP—should be mechanically resistant, chemically stable, and highly porous to minimize transfer resistance [62].

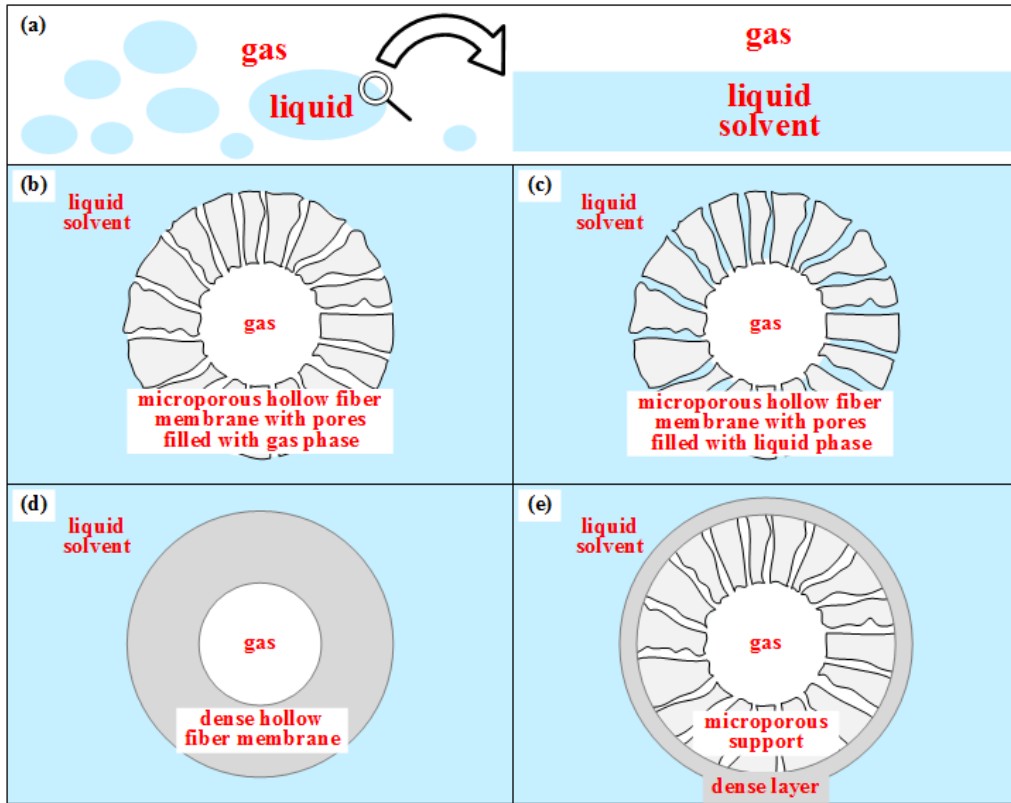

**Figure 3.** HFM as a barrier for mass transfer without phase dispersion: (**a**) direct *V-L* contact with phase dispersion; (**b**) microporous HFM *V-L* contact with gas-filled pores; (**c**) microporous HFM *V-L* contact with liquid-filled pores; (**d**) Dense HFM without *V-L* contact; (**e**) Composite HFM without *V-L* contact.

### 2.2. Solvents for GLMC CO₂ Absorption

Hereafter, physical and chemical GLMC solvents are discussed.

### 2.2.1. Physical Solvents

PhA relies on $CO_2$ solubility, which follows Henry's law, such that the absorption loading is favored by high $CO_2$ fugacity and low temperature. Thus, PhA can perform $CO_2$ removal from high-pressure raw NG. Comparatively to ChA, the main PhA advantage is the lower heat ratio for solvent regeneration [64]. Physical solvents are listed in Table 4 for GLMC $CO_2$ removal from NG.

**Table 4.** Physical solvents proposed for GLMC $CO_2$ removal from NG.

| Physical Solvent | Reference |
|---|---|
| Water | [65] |
| Propylene carbonate (Fluor$^{TM}$ process) | [66] |
| Dimethyl ethers of propylene glycol (DEPG) (Selexol$^{TM}$ process) | [67] |
| Chilled methanol (Rectisol$^{TM}$ process) | [68] |
| N-methyl-2-pyrrolidone (NMP) (Purisol$^{TM}$ process) | [26] |
| Ionic-liquid | [69] |

Some promising GLMC physical solvents are ionic liquids, which are salts of organic cations with organic anions. Ionic liquids exhibit melting points below 100 °C, negligible vapor pressure, high $CO_2$ solubility, and thermal stability but present high viscosity as a disadvantage [70].

### 2.2.2. Chemical Solvents

ChA is characterized by high $CO_2/CH_4$ selectivity and limited absorption loadings ($kg^{CO2}/kg^{Solvent}$), both due to chemical reactions of the solvent with $CO_2$. Aqueous-amine solvents are the most used in ChA for $CO_2$ capture from NG due to high reactivity, low cost, easy regeneration and high $CO_2/CH_4$ selectivity [71]. Aqueous-MEA, aqueous-DEA (aqueous-diethanolamine) and aqueous-MDEA are examples of aqueous-amine solvents extensively employed for NG decarbonation since 1930 [72]. However, aqueous-amines exhibit drawbacks such as high heat-ratio ($kJ/kg^{CO2}$) solvent regeneration, thermal/oxidative degradation, corrosiveness and limited $CO_2$ loading [73].

Aqueous-MEA is the most used for ChA $CO_2$ capture, exhibiting high reactivity at low-pressure or high-pressure [74], while aqueous-MDEA is adequate at high pressure despite lower absorption rate [75], which recommends using reactivity promoters. Table 5 lists literature studies on chemical solvents for GLMC NG decarbonation.

**Table 5.** Chemical solvents investigated for GLMC $CO_2$ removal from NG.

| Chemical Solvents | Reference |
|---|---|
| **Primary amines** | |
| Aqueous-MEA | [76] |
| Aqueous-DGA (Aqueous-Diglycolamine) | [77] |
| **Secondary amines** | |
| Aqueous-DEA | [78] |
| Aqueous-DIPA (Aqueous-Diisopropanolamine) | [79] |
| Aqueous-PZ (Aqueous-Piperazine) | [80] |
| Aqueous-EMEA (Aqueous-N-ethyl-monoethanolamine) | [81] |
| **Tertiary amines** | |
| Aqueous-TEA (Aqueous-Triethanolamine) | [80] |
| Aqueous-MDEA | [82] |
| **Sterically hindered amines** | |
| Aqueous-AMP (Aqueous-2-amino-2-methyl-1-propanol) | [83] |
| **Alkali-salts** | |
| Aqueous-$K_2CO_3$ | [84] |
| **Alkali** | |
| Aqueous-NaOH | [85] |
| **Amino-acid salts** | |
| Aqueous-AAS | [86] |

2.2.3. Solvent Blends

The literature presents $CO_2$ capture studies (for GLMC or not) with several blended solvents [87], namely: (i) aqueous-MDEA-PZ (PZ promoter) [88]; (ii) aqueous-MEA-MDEA, aqueous-MEA-DIPA and several combinations of aqueous-amines [89]; (iii) aqueous-PZ-$K_2CO_3$ (PZ promoter) [90]. Physical and chemical solvents are also blended, such as NMP with aqueous-ethanolamine [26] and ionic-liquid with aqueous-ethanolamine [91].

## 3. Experimental Studies: GLMC NG Decarbonation

Experimental studies exclusively on NG decarbonation involving GLMC-absorber and GLMC-stripper are reviewed, while studies on GLMC modeling from experiments are left to Section 4.

### 3.1. Experiments on GLMC $CO_2$ Absorption from NG

Qi and Cussler [92] started an investigation of HFM-GLMC in 1985 via experiments with $CO_2$ transfer in the liquid, followed by gas-to-liquid transfer across HFM [93]. They unveiled the advantages of HFM-GLMC over packing columns: the higher GLMC transfer area overcomes the HFM mass transfer resistance.

Dindore et al. [94] screened combinations of HFM and physical solvents for GLMC $CO_2$ capture by testing eight physical solvents with five HFM materials. Since few combinations were compatible, the authors demonstrated the importance of HFM-solvent compatibility to avoid MPW and its higher mass transfer resistance. The authors suggested the combination of propylene carbonate solvent with microporous PTFE-HFM or microporous PP-HFM, highlighting the high PTFE cost.

Atchariyawut et al. [95] studied $CO_2$ separation from $CO_2$+$CH_4$ gas mixtures in the GLMC shell-side with several solvents—water, aqueous-NaOH-NaCl, and aqueous-MEA— flowing inside HFM, evincing superiority of counter-current contact over parallel contact. Low $CO_2$/$CH_4$ selectivity was found and justified by the small GLMC module used. The operating pressure was not clearly informed, but it seems that low pressures were used.

Hedayat et al. [96] investigated GLMC for $H_2S$ and $CO_2$ capture from NG with aqueous-MDEA, aqueous-DEA-MDEA, and aqueous-MEA-MDEA solvents. The blend aqueous-DEA-MDEA did not provide any benefit over aqueous-MDEA, and it was observed that MEA preferably reacts with $CO_2$ and MDEA with $H_2S$. The best capture efficiencies were obtained with the aqueous-MEA-MDEA blend. The experiments were conducted at low pressures and are not representative of real NG processing conditions.

Al-Marzouqi et al. [97] experimentally evaluated different HFMs and solvents for $CO_2$ removal from $CO_2$+$CH_4$ mixtures with $CO_2$ = 10 mol%. Among the tested HFM, PP-HFM reached the best performance, while among the tested aqueous-amines, aqueous-TEPA (aqueous-tetraethylenepentamine), aqueous-TETA (aqueous-triethylenetetramine) and aqueous-DETA (aqueous-diethylenetriamine) reached better $CO_2$ removal rates comparatively to aqueous-MEA, aqueous-DEA, and aqueous-EDA (aqueous-ethylenediamine). Aqueous-NaOH presented higher $CO_2$/$CH_4$ selectivity than aqueous-MEA. Again, experiments were conducted at low pressure.

Marzouk et al. [98] was the first work with high-pressure (P = 50 bar) GLMC experiments on $CO_2$ removal from NG ($CO_2$ = 9.5 mol%, $CH_4$ = 90.5 mol%). Expanded polytetrafluoroethylene (ePTFE) microporous HFM was used and physical and chemical solvents—water, aqueous-NaOH, and aqueous-MEA—were evaluated. Results showed that the positive effect of increasing gas pressure is much more pronounced in physical solvents than in chemical solvents; i.e., physical solvents are suitable at high-pressure, while chemical solvents can work at low-pressure. Despite the need for investigations on $H_2S$ removal and simultaneous $CO_2$-$H_2S$ removal from NG at high pressure, experiments considered only $CO_2$ removal due to the risks of $H_2S$ experiments. Soon after, Marzouk et al. [99] modified the experimental setup to investigate high-pressure (P = 50 bar) $H_2S$ removal from NG ($H_2S$ = 2 mol%, $CH_4$ = 98 mol%) using the same HFMs and solvents from [98], finding $H_2S$ removal results similar to the $CO_2$ removal results of [98]. Posteri-

orly, Marzouk et al. [100] investigated GLMC simultaneous $CO_2$-$H_2S$ removal from NG ($CO_2$ = 5 mol%, $H_2S$ = 2 mol%, $CH_4$ = 93 mol%) at high-pressure (P = 50 bar) with same solvents of [98], and comparing ePTFE and PFA (poly(tetrafluoroethylene-co-perfluorinated alkyl vinyl ether)) as HFM. PFA-HFM presented $CO_2$ and $H_2S$ transmembrane fluxes ≈10 times higher than ePTFE-HFM counterparts, entailing transfer area reductions of 10 times in PFA-GLMC modules.

Rahim et al. [86] compared GLMC $CO_2$ absorption/stripping performances of four amino-acid salts (AAS) aqueous solvents with aqueous-MEA and aqueous-NaOH, showing that aqueous-AAS had the best $CO_2$ stripping performance as it just depends on pH changes. Albeit aqueous-NaOH reached the best absorption results, this solvent cannot be regenerated via ordinary $CO_2$ stripping, while aqueous-MEA can be regenerated but presented the worst $CO_2$ removal performance. However, all solvents were tested at 0.5 mol/L, while it is well-known that aqueous-MEA is better at 30 w/w% or ≈5 mol/L (i.e., ten-fold higher MEA composition). Apparently, the authors used very diluted aqueous-MEA to "establish" the superiority of aqueous-AAS.

Al-Marzouqi et al. [101] performed GLMC experiments to explore operating conditions for $CO_2$ capture from NG ($CO_2$ = 5 mol%, $CH_4$ = 95 mol%) at high-pressure (P = 50 bar) considering PFA-HFM and aqueous-NaOH, aqueous-DEA and aqueous-$K_2CO_3$ in long-term bench-scale runs. Pilot-scale tests were recommended.

Afterward, Hatab et al. [85] presented a contribution to the literature as, for the first time, spherical glass beads were investigated as additional packing in the solvent shell-side to create turbulence improving mass transfer. GLMC experiments were carried out for $CO_2$ removal from NG ($CO_2$ = 5 mol%, $CH_4$ = 95 mol%) using aqueous-NaOH and aqueous-amine solvents at pressures up to P = 25 bar. This work attracted attention as it proved the role of baffles in GLMC shell-side, an aspect frequently ignored. Transverse baffles can also lead to GLMC intensification.

Kang et al. [88] investigated GLMC $CO_2$ removal from $CO_2$-rich NG (45 mol%–70 mol% $CO_2$) at high-pressure (1 bar–60 bar) with PVDF-HFM and aqueous-MDEA-PZ as solvent. The $CO_2$ removal efficiency was enhanced by increasing pressure or transfer area or reducing feed flowrate at the expense of increasing the $CH_4$ loss rate.

Kartohardjono et al. [102] evaluated polyethyleneglycol-300 solvent for GLMC $CO_2$ capture from $CO_2$-$CH_4$ gas mixture ($CO_2$ = 30 mol%, $CH_4$ = 70 mol%). Albeit the solvent is not well-known, operational conditions were badly informed, and only obvious conclusions were drawn without relevant novelties.

The novelty in Ghobadi et al. [103] is the demonstration that smaller HFM diameters favor GLMC $CO_2$ separation. Low-pressure experiments were conducted with aqueous-MEA, aqueous-DEA, aqueous-TEA, and water solvents for $CO_2$ removal from the $CO_2$-$CH_4$ gas mixture ($CO_2$ = 2.5 mol%, $CH_4$ = 97.5 mol%). Aqueous-MEA was considered the best solvent.

Chan et al. [104] questionably affirmed that GLMC is disadvantageous in comparison with packing columns for NG processing due to the higher GLMC solvent/gas ratio for $CO_2$ removal. Works were cited to justify this claim in which low $CO_2$ loadings were obtained with aqueous-amine solvents. However, Chan et al. [104] eventually discovered that just increasing GLMC module length (i.e., increasing transfer area) proved to be sufficient to solve the problem.

Petukhov et al. [105] reviewed GLMC $CO_2$/$H_2S$ removal from NG with PP-HFM and aqueous-NaOH as solvent. Experiments were also conducted at a low-pressure range (1.5–8 bar) for $CO_2$/$H_2S$ removal from $CH_4$-$CO_2$-$H_2S$ gas feeds, which suggests that this study contemplates biogas purification. However, attention was not given to ChA thermal effects.

Hosseini et al. [106] proposed mass transfer intensification by applying vibration to GLMC to induce turbulence. Experiments at P = 1.5 bar were conducted with pure $CO_2$ and aqueous potassium glycinate 15 w/w% as solvent.

### 3.2. Experiments on GLMC $CO_2$ Stripping

The closed-loop combination of GLMC-absorber for $CO_2$ removal from NG with GLMC-stripper for $CO_2$ stripping is a low-weight and low-footprint promising solution for offshore NG processing. Mansourizadeh [107] conducted experiments on GLMC $CO_2$ absorption/stripping using water solvent in a closed loop, with pure $CO_2$ feed and pure $N_2$ stripping gas.

GLMC $CO_2$ stripping from ionic-liquid solvent has also been studied. Lu et al. [108] studied aqueous ionic-liquid solvents 1-butyl-3-methyl-imidazolium tetrafluoroborate ([bmin][BF$_4$]) and 1-(3-aminopropyl)-3-methyl-imidazolium tetrafluoroborate ([apmin][BF$_4$]) for coupled GLMC $CO_2$ absorption and GLMC solvent regeneration applying vacuum for $CO_2$ stripping. Aqueous-[bmin][BF$_4$] was completely regenerated, while aqueous-[apmin][BF$_4$] presented a difficult regeneration. Mulukutla et al. [109] investigated ionic-liquid [bmin][DCA] with 20 w/w% polyamidoamine for GLMC $CO_2$ capture and GLMC solvent regeneration showing that the stripped gas attained 92 mol% $CO_2$. Bazhenov et al. [110] proved the concept of high-pressure $CO_2$ stripping from ionic-liquid [Emin][BF$_4$] via moderate pressure-swing ($\Delta P = 10$ bar) coupled to temperature-swing. [Emin][BF$_4$] is recommended for pre-combustion $CO_2$ removal and NG sweetening with GLMC absorption/stripping. Simons et al. [111] described an experimental setup for testing ionic-liquid solvents for GLMC $CO_2$ absorption/stripping and demonstrated it with 50 w/w% aqueous dimethyl-propylenediamine acetates.

The majority of experiments on GLMC $CO_2$ stripping were conducted with aqueous-MEA since it is the most used solvent for GLMC $CO_2$ removal from $CO_2$-rich NG. Koon-aphapdeelert et al. [112] investigated $CO_2$ stripping from aqueous-MEA with ceramic HFM GLMC as this operation is performed preheating $CO_2$-rich aqueous-MEA at T = 100 °C, at which most polymeric HFM cannot withstand. Khaisri et al. [113] employed PTFE-HFM at the same conditions as [112], observing that increasing temperature and MEA content increase $CO_2$ stripping, but too high MEA contents decrease $CO_2$ transfer due to high viscosity. Khaisri et al. [114] presented a similar work using a similar GLMC-stripper experimental setup. Scholes et al. [115] tested three HFM at a temperature range of 70 °C–105 °C, confirming higher $CO_2$ stripping fluxes at higher temperatures and higher Reynolds numbers as turbulence decreases boundary layer mass transfer resistance. Scholes et al. [54] tested steam as a stripping agent for $CO_2$ stripping from aqueous-MEA. Kianfar et al. [116] investigated polysulfone (PSF) as HFM for GLMC-absorber and GLMC-stripper. $CO_2$ stripping was performed at T = 80 °C and P = 0.5 bar with a pure $N_2$ stripping agent. Lee et al. [117] developed a vacuum-driven stripping method at P = 61.325 kPa. Comparatively to the common $N_2$ sweep-gas stripping, this process reduces heat consumption, increasing stripping efficiency and $CO_2$ purity due to $N_2$ absence.

Experimental studies of $CO_2$ stripping from aqueous-DEA focused on HFM development and adopted $N_2$ as stripping gas at T = 80 °C. Naim et al. [118] prepared different PVDF-HFM with LiCl as a non-solvent additive. It was observed that increasing HFM LiCl content increases mass transfer performance. Naim et al. [119] tested LiCl, glycerol, polyethylene-glycol (PEG400), methanol, and H$_3$PO$_4$ non-solvent additives to PVDF-HFM, showing that PVDF-PEG400-HFM presented the highest $CO_2$ transfer flux. Naim and Ismail [120] confirmed the potential of polyetherimide (PEI) HFM, while Naim et al. [121] compared PVDF-HFM with PEI-HFM. Both were prepared with polyethylene-glycol (PEG) incorporation and PVDF-PEG-HFM presented a higher $CO_2$ transfer flux. Rahbari-Sisakht et al. [122] developed a PVDF-HFM with surface modifying molecules. Hosseini and Mansourizadeh [123] investigated microporous hydrophobic poly(vinylidene fluoride-co-hexafluoropropylene) (PVDF-HFP) HFM.

Aqueous-MDEA-PZ for GLMC $CO_2$ absorption/stripping was studied by Akan et al. [124], showing that vacuum-driven $CO_2$ stripping was achieved at T = 92 °C using helium as stripping gas. Rahim et al. [125] evaluated PVDF-HFM GLMC $CO_2$ stripping from different solvents: aqueous-MEA, aqueous-DEA, aqueous-AMP, and aqueous potassium glycinate. For all solvents, the authors demonstrated that mass transfer increases by

increasing temperature and initial $CO_2$ content. Wang et al. [126] investigated aqueous solutions of 16 amines via fast screening experiments to find the solvents with lower heat-ratio for $CO_2$ stripping in a combined GLMC absorption/stripping process. In terms of overall performance, aqueous-MDEA was recommended for GLMC-stripper.

Finally, Kim et al. [127] investigated the microporous and hydrophobic electrospun poly(vinylidene fluoride-co-hexafluoro propylene) nanofiber membrane for $CO_2$ stripping from aqueous potassium glycinate.

## 4. Modeling Studies: GLMC NG Decarbonation

The development of accurate GLMC modeling is of great importance as it avoids spending money and time on experiments as well as providing the necessary information to predict and analyze GLMC performance [128]. This section presents an extensive critical review of GLMC modeling for $CO_2$ removal from NG and for $CO_2$ stripping. Throughout Section 4, Tables 6–8 are cited in order to present mathematical relationships related to the works and models under discussion. As Tables 6–8 naturally belong to the context of Section 4.3, they are installed there.

### 4.1. GLMC Modeling for $CO_2$ Absorption from NG

As NG decarbonation is performed at high-pressure and frequently with concentrated aqueous-amine solvents highly deviating from ideal solution behavior, rigorous thermodynamic models must be used for property prediction, comprising an Equation-of-State (EOS) for the vapor phase and an activity coefficient model for the liquid phase. Moreover, mass, heat, and momentum transfers occur simultaneously and influence each other; i.e., related predictions must be accurate to ensure reliability in industrial scale conditions. Moreover, the multicomponent mass transfer must be considered as $CO_2/H_2S$ simultaneous removal is necessarily entailing unavoidable transfers of $CH_4$ and other hydrocarbons. Besides, $H_2O$ can move in both directions, $V{\rightarrow}L$ or $L{\rightarrow}V$.

Dindore et al. [129] modeled $CO_2/H_2S$ mass transfer with chemical reaction in a cross-flow GLMC with aqueous-$K_2CO_3$ solvent. The model is simplified and does not include energy balance and thermal effects. Experiments were carried out with a $CO_2$-$H_2S$-$N_2$ gas mixture which deviates from NG behavior as $CH_4$ is considerably more soluble than $N_2$. Therefore $CO_2/CH_4$ selectivity and $CH_4$ losses were not evaluated.

The GLMC models for $CO_2$ removal from NG of Al-Marzouqi et al. [65] (Table 7, Equations (47), (48), (50), (51), (53), (55) and (56)) with water solvent and Al-Marzouqi et al. [130] (Table 7, Equations (47), (48), (50), (51), (53), (55) and (56)) with aqueous-MEA and aqueous-NaOH were developed for low-pressure GLMC. Faiz and Al-Marzouqi [131] presented a low-pressure GLMC model (Table 7, Equations (47), (50) and (55)) based on reaction kinetics for $H_2S$ absorption in aqueous-$Na_2CO_3$, which is not suitable for high-pressure NG processing. Furthermore, NG purification requires $CO_2/H_2S$ simultaneous removal. Faiz and Al-Marzouqi [76] developed a low-pressure model (Table 7, Equations (47), (48), (50), (51), (53), (55)–(60)) for $CO_2/H_2S$ simultaneous removal with aqueous-MEA. Keshavarz et al. [78] presented a simplified GLMC model (Table 7, Equation (55), Table 8, Equation (68)) for $CO_2/H_2S$ simultaneous removal from $CO_2$-$H_2S$-$N_2$ gas mixture ($CO_2$ = 15 mol%, $H_2S$ = 5 mol%, $N_2$ = 80 mol%) with aqueous-DEA. Unfortunately, low-pressure and isothermal behavior was considered an unrealistic situation in high-pressure $CO_2/H_2S$ absorption from NG with aqueous-amines, which is a highly exothermic process. In addition, ideal gas behavior was also assumed.

Faiz and Al-Marzouqi [132] developed the first GLMC modeling study at high-pressure (P ≤ 50 bar) for $CO_2$ removal from NG considering water and aqueous-amines solvents. The model (Table 7, Equations (47), (48), (50), (51), (53), and (55)–(57)) consisted of differential equations in cylindrical coordinates for $CO_2$ mass transfer solved with COMSOL software and was validated with Marzouk et al. [98] experiments. GLMC transfers of $H_2S$, $CH_4$, and water were not taken into account. The authors justified neglecting $CH_4$ transfer due to the low $CH_4$ solubility in aqueous solutions. However, this assumption is

problematic because with physical solvents at high-pressure, $CH_4$ losses can reach 9–10%, while with chemical solvents, $CH_4$ losses are considerably smaller but never negligible. In addition, this model does not prescribe energy balance and head-loss calculations; i.e., heat effects and temperature/pressure changes were not predicted. Posteriorly, Faiz and Al-Marzouqi [133] presented a low-pressure isothermal GLMC model (Table 7, Equations (47), (48), (50), (51), (53), (55), and (56)) for $CO_2/H_2S$ simultaneous removal from NG with aqueous-$K_2CO_3$ solvent. The authors implemented two serial GLMC modules to favor reaction kinetics so that all the $H_2S$ and a small portion of $CO_2$ are removed in module #1 and the remaining $CO_2$ is removed in module #2. However, this low-pressure isothermal model is not compatible with real NG processing conditions.

Faiz et al. [134] proposed a GLMC model (Table 7, Equations (47), (48), (50), (51), (53), and (55)–(57)) taking into account the gas velocity change in the axial direction as $CO_2$ is removed from the gas initially with 10 mol% $CO_2$. The model was validated with experiments of [97] and showed, for high $CO_2$ content in the gas feed, that the effect of gas velocity change is more evident since the gas velocity decreases significantly. The decrease in gas velocity implies a higher residence time, entailing higher $CO_2$ capture. Model predictions were in better agreement with the experimental data for chemical solvents, but they were not good for physical solvents. This study also contemplates a low-pressure GLMC NG purification.

Rezakazemi et al. [82] studied $CO_2/H_2S$ simultaneous GLMC removal from NG using aqueous-MDEA as solvent. A two-dimensional model was proposed (Table 7, Equations (48), (51)–(53), (55), and (56)), considering axial and radial diffusion, convection, chemical reactions, and momentum equations. The model was solved with CFD techniques. However, the model was developed at low-pressure and ambient temperature, entailing oversimplified assumptions; namely: (i) isothermal GLMC profiles are unrealistic in a system with exothermic reactions of $CO_2$ with concentrated aqueous-MDEA; (ii) ideal gas behavior is not adequate at real operating conditions of high-pressure NG sweetening. The authors affirm that $CH_4$ does not dissolve in the solvent; another unrealistic outcome and water mass transfer was also ignored.

Boributh et al. [135] developed a 1D GLMC model (Table 6, Equation (37)) with water solvent for estimating the mass transfer area considering different GLMC module arrangements: single-stage module, two-stage parallel modules, two-stage serial modules with the separated liquid flow, and two-stage serial modules with combined liquid flow. In Boributh et al. [136] (Table 6, Equation (37)), the GLMC with water solvent model includes the effect of pore size distribution to predict MPW and CO2 mass transfer correctly. Boributh et al. [137] proposed a GLMC model (Table 6, Equation (37)) for the design of multistage GLMC schemes with aqueous-MEA solvent. Works [135–137] modeled countercurrent GLMC $CO_2$ removal from $CO_2$-$CH_4$ gas mixtures considering partial MPW. These works are similar, performed low-pressure case studies, and presented the same GLMC modeling simplifications: isothermal conditions, ideal gas behavior, and zero head-loss.

A new GLMC concept is found in Cai et al. [138], which proposed to carry out both the acid–gas absorption and solvent regeneration in a single step, i.e., in the same GLMC cross-flow modules. The aim is to save space in offshore processing plants. In this concept, the solvent flows on the shell-side, while two types of HFM are used in the module. $CO_2$-rich NG flows in the microporous HFM at high-pressure (P = 50 bar), wherein $CO_2/H_2S$ absorptions take place at the gas-liquid interface. Inside the GLMC module, there is also dense HFM, kept at low-pressure (P = 1 bar) so that the difference of pressure in relation to the solvent generates a stripping driving force which promotes solvent regeneration in the shell-side as $CO_2/H_2S$ scape to the low-pressure HFM. Therefore, $CO_2/H_2S$ absorption and solvent regeneration occur simultaneously in the GLMC module, theoretically resulting in better absorption efficiency as the solvent is kept with low acid-gas content while flowing through the GLMC module. The authors also suggest using baffles to improve mass transfer efficiency by increasing turbulence. Propylene carbonate was the physical solvent as it presents better $CO_2$ solubility in comparison with water. The use of physical solvents

is recommended as chemical solvents demand heat-intensive regeneration and require more space on offshore rigs. Indeed, it is easy to understand that this GLMC concept would never work with chemical solvents because chemical solvents would necessarily require a GLMC-absorber and a separate solvent regeneration step. The GLMC model considers isothermal mass transfers through the microporous HFMs, through the dense HFMs, and on the shell side. The authors assumed isothermal conditions and neglected gas head-losses. Despite the new concept, this work did not trigger analogous works by other authors since then.

An important and disruptive contribution comes from Ghasem et al. [60], as it was the first model to consider GLMC water vaporization and the respective heat effect, a phenomenon systematically ignored in the GLMC modeling literature, which prefers to assume isothermal GLMC modeling without water mass transfer, a fact that Ghasem et al. proved to be unrealistic. In [60], GLMC $CO_2$ absorption from NG was performed using an aqueous-NaOH solvent. In comparison with previous works, a more complete GLMC model was developed (Table 7, Equations (47), (49)–(51), (54), (56), and (61)–(67)). Instead of considering isothermal GLMC and neglecting pressure drop, the model [60] considers mass, energy, and momentum balances. The work carried out experiments and results were compared with model predictions. Nonetheless, experiments were performed at atmospheric pressure with a raw NG composition of 10 mol% $CO_2$ and 90 mol% $CH_4$. Besides, in real NG processing, the high-pressure raw NG is water-saturated, so that water can transfer in both directions, a fact not considered in [60]. Moreover, real raw NG contains $H_2S$ and light hydrocarbons ($C_2^+$).

After the good work in [60], Ghasem et al. [139] published a badly written article with several typos, including the article title and the titles of some sections. $CO_2$ absorption from NG was investigated experimentally and theoretically with a GLMC model comprising only material balances (Table 7, Equations (47), (48), (51), (53), and (56)). Aqueous potassium glycinate was compared with aqueous-amine solvents showing superior results, a questionable outcome. The article is a mess: the text creates confusion among MEA, DEA, and MDEA all the time. Thereby, it is impossible to be sure of which aqueous-amine solvents were investigated.

A complete GLMC thermodynamic model is found in de Medeiros et al. [33], which simulated an offshore high-pressure NG purification process for GLMC $CO_2$ removal with aqueous-MEA-MDEA. The GLMC model comprises a new acid-gas, water, and alkanolamines equilibrium model (AGWA model) [140] capable of reproducing chemical absorption and desorption of $CO_2$ (and $H_2S$) in aqueous-amines, over a wide range of pressures, which was calibrated in terms of equilibrium constants with AGWA VLE data from the literature. The real species VLE is superimposed on the multi-reactive chemical equilibrium for the formation of fictitious species (non-ionic and non-volatile complexes), defining a reactive VLE or RVLE [140]. The GLMC *V* and *L* flows are parallel and unidimensional (1D), assuming compressible two-phase plug-flow for *V* and *L* streams. The model also considered the mass transfer of $CH_4$, $H_2O$, and light hydrocarbons and was developed through rigorous mass, energy, and momentum balances for *V* and *L* streams, allowing heat and mass interfacial transfers, with shear and compressibility terms, and all properties of all phases rigorously calculated by the Peng–Robinson EOS (PR-EOS) and Soave–Redlich–Kwong EOS (SRK-EOS). This model was able to reproduce several interesting behaviors along the GLMC length as profiles of both streams, such as molar flowrates, species compositions, temperatures, pressures, fugacities, and $CO_2/CH_4$ selectivity [33]. Model equations are shown in Equations (1) to (7).

**Trans-membrane fluxes of real species "*k*":**

$$N_k^{flux} = \Pi_k * \left( \hat{f}_{V_k} - \hat{f}_{L_k} \right) \qquad (1)$$

**Real species EMWR (Equivalent Moles Without Reaction) mass balances for *V* and *L* streams:**

$$\frac{dV_k}{dz} = -Sa * N_k^{flux} \tag{2}$$

$$\frac{dL_k}{dz} = Sa * N_k^{flux} \tag{3}$$

where $N_k^{flux}$ are trans-membrane fluxes of real species "$k$"; $\Pi_k$ are trans-membrane mass transfer coefficients of species "$k$"; $\hat{f}_{V_k}$ and $\hat{f}_{L_k}$ are the $V$ and $L$ fugacities of real species "$k$"; $Sa$ is the interfacial area per length; $V_k$ and $L_k$ are $V$ molar flowrate and the $L$ EMWR molar flowrate of real species "$k$."

**Momentum balances for *V* and *L*:**

$$\left\{ \Gamma_{T_V} * (v_V)^2 \right\}\frac{dT_V}{dz} + \left\{ \Gamma_{P_V} * (v_V)^2 - 1 \right\}\frac{dP_V}{dz} + \sum_{k}^{n_{RS}} \left\{ \Gamma_{V_k} * (v_V)^2 \right\}\frac{dV_k}{dz} = \rho_V g * sen\theta - \left(\frac{v_V}{S_V}\right)qSa + \psi_V\left(\frac{\Pi D + Sa}{S_V}\right) \tag{4}$$

$$\left\{ \Gamma_{T_L} * (v_L)^2 \right\}\frac{dT_L}{dz} + \left\{ \Gamma_{P_L} * (v_L)^2 - 1 \right\}\frac{dP_L}{dz} + \sum_{k}^{n_{RS}} \left\{ \Gamma_{L_k} * (v_L)^2 \right\}\frac{dL_k}{dz} = \rho_L g * sen\theta + \left(\frac{v_L}{S_L}\right)qSa$$
$$+ \left(\frac{v_L - v_V}{S_L}\right)qSa + \psi_L\left(\frac{Sa}{S_L}\right) \tag{5}$$

where $D$ is the GLMC shell internal diameter, $\theta$ is GLMC inclination, $g$ is the gravity acceleration, $n_{RS}$ is the number of real species, $\Gamma_{T_V}$ and $\Gamma_{T_L}$ are isobaric expansivity of $V$ and $L$ at RVLE, $\Gamma_{P_V}$ and $\Gamma_{P_L}$ are isothermal compressibility of $V$ and $L$ at RVLE, $\Gamma_{V_k}$ and $\Gamma_{L_k}$ are species "k" mol expansivity of $V$ and $L$ at RVLE, $T_V$ and $T_L$ are $V$ and $L$ temperatures, $P_V$ and $P_L$ are $V$ and $L$ absolute pressures, $v_V$ and $v_L$ are $V$ and $L$ flow velocities, $\rho_V$ and $\rho_L$ are $V$ and $L$ densities, $S_V$ and $S_L$ are sections of $V$ and $L$ flows, $\psi_V$ and $\psi_L$ are the shear stress on contact surfaces in $V$ and $L$ flows.

**Energy balances for *V* and *L*:**

$$\left\{ \frac{\rho_V \overline{C}_{P_V}}{M_V} - \Gamma_{T_V}(v_V)^2 \right\}\frac{dT_V}{dz} + \left\{ 1 + \frac{T_V}{\rho_V}\Gamma_{T_V} - \Gamma_{P_V}(v_V)^2 \right\}\frac{dP_V}{dz} - \sum_{k}^{n_{RS}} \left\{ \Gamma_{V_k} * (v_V)^2 \right\}\frac{dV_k}{dz} =$$
$$\left(\frac{v_V}{S_V}\right)qSa - \rho_V g * sen\theta + \frac{\Omega_E \Pi D \rho_V}{q_V}\left(T^{amb} - T_V\right) - \left(\frac{\Omega_I Sa \rho_V}{q_V}\right)(T_V - T_L) \tag{6}$$

$$\left\{ \frac{\rho_L \overline{C}_{P_L}}{M_L} - \Gamma_{T_L}(v_L)^2 \right\}\frac{dT_L}{dz} + \left\{ 1 + \frac{T_L}{\rho_L}\Gamma_{T_L} - \Gamma_{P_L}(v_L)^2 \right\}\frac{dP_L}{dz} - \sum_{k}^{n_{RS}} \left\{ \Gamma_{L_k} * (v_L)^2 \right\}\frac{dL_k}{dz} =$$
$$-\left(\frac{v_L}{S_L}\right)qSa - \rho_L g * sen\theta + \left(\frac{\Omega_I Sa \rho_L}{q_L}\right)(T_V - T_L) + \left(\frac{Sa \rho_L}{q_L}\right)\sum_{k}^{n_{RS}} N_k^{flux}\left(\overline{E}_{V_k} - \overline{E}_{L_k}^{EMWR}\right) \tag{7}$$

where $\overline{C}_{P_V}$ and $\overline{C}_{P_L}$ are $V$ and $L$ molar isobaric heat capacities, $M_V$ and $M_L$ are the $V$ and $L$ EMWR molar masses, $q_V$ and $q_L$ are $V$ and $L$ mass flowrates, $q$ is the HFM mass flux, $\Omega_E$ and $\Omega_I$ are external and internal heat transfer coefficients, $T^{amb}$ is the outside temperature, $\overline{E}_{V_k}$ and $\overline{E}_{L_k}^{EMWR}$ are the PMP$^{EMWR}$ energies of "$k$" in $V$ and $L$.

Faiz et al. [141] described a 2D GLMC model (Table 7, Equations (48), (50), (51), (53), and (55)–(57)) for $H_2S$ removal from high-pressure NG with water solvent. Model predictions agreed perfectly with the experimental data only at low pressures. At high pressures, agreement with experimental data only occurred when the "pseudo-wetting" condition (3%) (MPW) was assumed. The model ignores the transfer of other species, energy balances, and head losses.

Razavi et al. [79] modeled low-pressure GLMC $CO_2$ removal from NG with an aqueous di-isopropanol amine (DIPA) solvent. The 2D GLMC model (Table 7, Equations (47), (48), (50)–(53), (55), and (56)) ignores thermal effects (isothermal GLMC), assumes ideal gas behavior, and considers only species mass balances without head-loss calculations.

Cai et al. [142] modeled crossflow GLMC for $CO_2$ removal from NG. Experiments were performed and a CFD modeling was solved with ANSYS software. The model considers

mass and momentum balances, without energy balance, in three-dimensional geometry. The study considered carbon capture from pure $CO_2$ with water solvent at P = 1 atm and T = 298 K, which are distant values from real NG processing conditions.

Abdolahi-Mansoorkhani and Seddighi [80] investigated HFM improvements with the incorporation of montmorillonite nanoparticles for $CO_2$ removal from $CO_2$-$CH_4$ mixtures ($CO_2$ = 10 mol%, $CH_4$ = 90 mol%) using aqueous-MEA, aqueous-PZ, aqueous-TEA, and aqueous-EDA solvents. A 2D finite element modeling in cylindrical coordinates predicted GLMC behavior (Table 7, Equations (47), (48), (50)–(53), and (55)–(60)). Despite the highly exothermic ChA with aqueous-amines, the model ignores energy balances, considering isothermal profiles without head-loss, ideal gas behavior, and VLE via Henry's law. Besides, only $CO_2$ is transferred.

Nakhjiri et al. [143] conducted experiments and developed a two-dimensional (2D) GLMC model (Table 7, Equations (47), (48), (50), (51), (53) and (55)–(57)) for prediction of $CO_2$ removal from $CO_2$-$CH_4$ mixtures ($CO_2$ = 30 mol%, $CH_4$ = 70 mol%) with aqueous-MEA and aqueous-TEA. Predicted results were compared with experimental counterparts in both non-wetting and partial wetting (MPW) modes of operation. Model differential equations were solved with COMSOL software. This work did not unveil any noticeable novelty, as the isothermal low-pressure 2D model was developed with simplifications found in previously published works such as ideal gas behavior, Henry's law for VLE, absence of $H_2S$ and $H_2O$ in raw NG, only $CO_2$ transference, and no head-loss. Nakhjiri et al. [144] used almost the same equations (Table 7, Equations (47), (48), (50), (51), (53), and (55)–(60)) of [143], with the inclusion of aqueous-NaOH experiments, finding the following order of $CO_2$ removal performances: aqueous-MEA > aqueous-NaOH > aqueous-TEA. Model predictions were compared with data from [95]. Nakhjiri and Heydarinasab [145] studied GLMC $CO_2$ removal from a $CO_2$-$CH_4$ gas mixture ($CO_2$ = 10 mol%, $CH_4$ = 90 mol%) with aqueous ethylenediamine (EDA), aqueous 2-(1-piperazinyl)-ethylamine (PZEA) and aqueous potassium sarcosinate (PS) solvents using a model (Table 7, Equations (47), (48), (50)–(53) and (55)–(57)) similar to [143,144] models, considering non-wetting mode of operation. Model predictions were compared with data from [65] for validation, an awkward procedure since the experiments from [65] were conducted with pure water solvent. Posteriorly, Nakhjiri et al. [84] slightly improved their previous GLMC model [143] by considering simultaneous $CO_2$/$H_2S$ removal with aqueous-$K_2CO_3$ solvent, including momentum balance equations (Table 7, Equations (47), (48), (50), (51), (53), (55)–(57) and (61)–(64)). However, all the remaining simplifications of [143–145] were kept.

Mesbah et al. [90] presented a theoretical study of GLMC $CO_2$ removal from $CO_2$-$CH_4$ gas mixtures using an aqueous-$K_2CO_3$ solvent containing a 2-methylpiperazine (2MPZ) promoter. Despite considering a complex chemical reaction scheme of $CO_2$ with aqueous-$K_2CO_3$-2MPZ, the 2D GLMC model (Table 7, Equations (47), (48), (50)–(53) and (55)–(57)) is simplified with the following assumptions: isothermal profiles, low-pressure, ideal gas behavior, Henry's law VLE, and only $CO_2$ is transferred. Awkwardly, the supposed model validation was performed with experimental data from [143], which involved only aqueous-MEA and aqueous-TEA solvents.

Fougerit et al. [146] performed an experimental and modeling study of GLMC $CO_2$ removal from $CO_2$-$CH_4$-$H_2O$ gas mixtures covering a wide range of $CO_2$ contents, namely: NG sweetening ($CO_2$ = 5–10 mol%), post-combustion carbon capture ($CO_2$ = 10–15 mol%), biogas upgrading ($CO_2$ = 35–50 mol%). However, the authors ignored that each feed mixture needs different process conditions. Besides, the 1D mass transfer model is simplified: only $CO_2$ transference is considered, isothermal GLMC, ideal gas behavior, and low-pressure (P = 5.0 bar).

The GLMC decarbonation of coal-bed methane, an NG abundant in China, was studied by Zhang et al. [147] with a 2D GLMC model (Table 7, Equations (48), (50), (51), (53) and (55)–(57)) considering aqueous-MEA, aqueous-DEA, aqueous-TEA and aqueous 1-(2-hydroxyethyl)pyrrolidine solvents. The GLMC model is also simplified: low-pressure, ideal gas behavior, and isothermal GLMC profiles were adopted for ChA with aqueous-amine

solvents. Pan et al. [148] adopted the same GLMC model (Table 7, Equations (48), (50), (51), (53), (55), and (56)) for the simultaneous $CO_2/H_2S$ removal from coal-bed methane using aqueous-$K_2CO_3$ with potassium lysinate (PL) solvent. The pressure influence in the range 1 bar–50 bar on $CO_2/H_2S$ transfers was studied (still under ideal gas behavior).

Ghasem [149] experimentally studied GLMC $CO_2$ absorption from $CO_2$-$CH_4$ mixtures ($CO_2$ = 10 mol%, $CH_4$ = 90 mol%) and also the $CO_2$ stripping from the solvent. The solvent was 0.5 mol/L aqueous potassium glycinate in a closed loop with $N_2$ as stripping gas. Experimental results were compared with predictions of a GLMC model, which is 2D axisymmetric and transient, considering $CO_2/CH_4$ transports in GLMC-absorber, and $CO_2/N_2$ transports in GLMC-stripper. The model is simplified: it considers low-pressure, ideal gas behavior, isothermal profiles, VLE via Henry's law, and lacks energy balance and head-loss calculations. $H_2S$ and $H_2O$ are absent. Ghasem [150] investigated the influence of HFM bore size on the highly exothermic $CO_2$ removal from NG with aqueous-MEA, considering VLE via Henry's law. In comparison with [149], it was used a similar low-pressure 2D GLMC model at steady-state (Table 7, Equations (47), (48), (50), (51), (53), (55) and (56)).

GLMC Modeling with Professional Process Simulators for $CO_2$ Removal from NG

Implementation of GLMC modeling for carbon capture from NG in professional process simulators is crucial for rigorous GLMC modeling due to more accurate thermodynamic properties estimation [128].

Among the few works implementing GLMC modeling in process simulators, there are studies on $CO_2$ removal from other sorts of gas streams (e.g., biogas and syngas). Kerber and Repke [151] studied GLMC biogas purification, while Villeneuve et al. [152,153] studied carbon capture from post-combustion flue gas with GLMC implementation in Aspen Custom Modeler. Usman et al. [154] developed a GLMC model for pre-combustion $CO_2$ capture in MATLAB. Posteriorly, Usman et al. [155] integrated the MATLAB code with the HYSYS process simulator through the Cape-Open simulation compiler.

There are just a few works on GLMC modeling in process simulators for $CO_2$ removal from NG. Hoff et al. [156] conducted laboratory-scale experiments on $CO_2$ removal from flue gas with aqueous-MEA and on high-pressure $CO_2$ removal from NG with aqueous-MDEA. The authors developed a 2D diffusion-reaction model to predict experimental absorption rates and concentration/temperature GLMC profiles. The model considers mass, energy, and momentum balances, and VLE was modeled via Henry's law with activity coefficients to account for liquid phase non-ideality. The model equations—shown in Equations (8) to (16)—were solved via MATLAB.

**Gas mass balance equations:**

$$\frac{dN_k^{flux}}{dz} = \frac{v_V}{RT_V}\frac{\partial P_V}{\partial z} + \frac{P_V}{RT_V}\frac{\partial v_V}{\partial z} - \frac{P_V v_V}{R(T_V)^2}\frac{\partial T_V}{\partial z} = \sum_k N_k^{flux}\frac{A_{SISA}}{\varepsilon_V} \qquad (8)$$

$$\frac{dv_V}{dz} = \left(-\frac{v_V}{P_V}\frac{\partial P_V}{\partial z}\right) + \left(\frac{v_V}{T_V}\frac{\partial T_V}{\partial z}\right) - \left(\frac{RT_V}{v_V}N_k^{flux}\frac{A_{SISA}}{\varepsilon_V}\right) \qquad (9)$$

$$\frac{dP_{V_k}}{dz} = \left(-\frac{P_{V_k}}{v_V}\frac{\partial v_V}{\partial z}\right) + \left(\frac{P_{V_k}}{T_V}\frac{\partial T_V}{\partial z}\right) - \left(\frac{RT_V}{v_V}N_k^{flux}\frac{A_{SISA}}{\varepsilon_V}\right) \qquad (10)$$

$$\sum_k \overline{C}_{P_{V,k}} N_k^{flux}\frac{\partial T_V}{\partial z} = Q \qquad (11)$$

where $R$ is the ideal gas constant, $A_{SISA}$ is the specific GLMC inner surface area, $\varepsilon_V$ is the fraction of total area available to gas flow, $\overline{C}_{P_{V,k}}$ is the $V$ molar isobaric heat capacity of species "$k$" and $Q$ is the heat rate.

**Trans-membrane mass flux of species "$k$":**

$$N_k^{flux} = \frac{\left(P_{V_k} - H_k C_{L_k,m}\right)}{RT_V \left(\frac{1}{k_{k,V}} + \frac{r_o \ln(r_o/r_i)}{D_{k,m}^{eff}}\right)} + X_k \sum_k N_k^{flux} \tag{12}$$

$$D_{k,m}^{eff} = \frac{D_k \varepsilon}{\tau} \tag{13}$$

$$Q = \frac{1}{\left(\frac{1}{\Omega_V} + \frac{1}{\Omega_m}\right)}(T_V - T_{L,m}) \tag{14}$$

where $P_{V_k}$ is the partial pressure of "$k$," $H_k$ is Henry's law constant of "$k$," $C_{L_k,m}$ is the concentration of "$k$" at the liquid–membrane interface, $k_{k,V}$ is the gas film mass transfer coefficient of "$k$," $r_i$ and $r_o$ are the internal and external HFM radiuses, $D_{k,m}^{eff}$ is the effective membrane diffusivity of "$k$," $D_k$ is the molecular diffusivity of "$k$," $\varepsilon$ is membrane porosity, $\tau$ is membrane tortuosity, $\Omega_V$ is $V$ heat transfer coefficient, $\Omega_m$ is the HFM heat transfer coefficient and $T_{L,m}$ is the temperature at liquid-membrane interface.

**Transport model for the liquid phase based upon mass and energy conservation:**

$$2v_L \left[1 - \left(\frac{r}{r_i}\right)^2\right] \frac{\partial C_{L_k}}{\partial z} = D_k \left(\frac{1}{r}\frac{\partial C_{L_k}}{\partial r} + \frac{\partial^2 C_{L_k}}{\partial r^2}\right) + \frac{\partial D_k}{\partial r}\frac{\partial C_{L_k}}{\partial r} + R_k \tag{15}$$

$$2v_L \left[1 - \left(\frac{r}{r_i}\right)^2\right] \overline{C}_{P_L} \rho_L \frac{\partial T_L}{\partial z} = \lambda_L \frac{1}{r}\frac{\partial}{\partial r}\left(r\frac{\partial T_L}{\partial r}\right) + R_k(-\Delta H_r) \tag{16}$$

where $C_{L_k}$ is $L$ molar concentration of "$k$," $R_k$ is the chemical reaction rate in terms of "$k$," $\overline{C}_{P_L}$ is the $L$ molar isobaric heat capacity, $\lambda_L$ is $L$ thermal conductivity, and $\Delta H_r$ is the enthalpy of reaction.

Posteriorly, the model of Hoff et al. [156] was improved for utilization in process simulators, including new thermodynamic models for aqueous-MEA and aqueous-MDEA/PZ, and solvent flow modeling inside HFM or in shell-side, besides improvements for gas and solvent head-loss calculation. In Hoff and Svendsen [157], the GLMC model comprises an RVLE $CO_2$-water-amines system based on the Deshmukh–Mather approach [158], employing a modified Debye–Hückel model similar to Kuranov et al. [159]. The gas phase was modeled with PR-EOS and the trans-membrane fluxes were modified as shown in Equations (17) to (19).

$$N_k^{flux} = \frac{\left(\hat{f}_{V_k} - \hat{f}_{L_k}\right)}{ZRT_V \left(\frac{1}{k_{k,V}} + \frac{1}{k_{k,m}} + \frac{H}{Ek_{k,L}}\right)} + X_k \sum_k N_k^{flux} \tag{17}$$

$$k_m = \frac{\varepsilon D_p}{\tau r_i \ln(r_o/r_i)} \tag{18}$$

$$D_p = \frac{1}{(1 - X_L)\left(\frac{1}{D_m} + \frac{1}{D_{Kn}}\right) + \left(X_L \frac{1}{D_{m,L}}\right)} \tag{19}$$

where $Z$, $k_{k,V}$, $k_m$, $k_{k,L}$, $E$, $D_p$, $D_m$, $D_{Kn}$, $D_{m,L}$, $X_L$ are respectively, compressibility factor, gas film, membrane and liquid film mass transfer coefficients of "$k$," enhancement factor, combined diffusivity, membrane diffusivity, Knudsen diffusivity, diffusivity of membrane at liquid phase and partial liquid penetration, respectively.

In Hoff and Svendsen [157], the following gas treatments were simulated: (i) low-pressure post-combustion carbon capture from offshore gas-turbines flue-gas to attain 90% capture with 30 w/w% aqueous-MEA in GLMC cross-flow modules installed in the process simulator SINTEF/NTNU CO2SIM; (ii) high-pressure (P = 70 bar) offshore sweetening of

NG ($CO_2$ = 10 mol%) to $CO_2$ = 2.5 mol% using 30 w/w% aqueous-MDEA in GLMC parallel-flow modules installed in the process simulator Protreat; (iii) high-pressure (P = 70 bar) offshore sweetening of NG ($CO_2$ = 10 mol%) to LNG specification of $CO_2$ = 50 ppm-mol using 30 w/w% aqueous-MDEA with 5 w/w% PZ in GLMC parallel-flow modules installed in the process simulator Protreat. In comparison with packing columns, total equipment volume can be reduced to 25% using GLMC with liquid on the shell side. With liquid flowing inside HFM, the advantage reduces significantly. Authors highlighted that GLMC may significantly improve offshore $CO_2$ capture from gas–turbine flue-gas and for NG sweetening but pointed out the doubtful advantage of GLMC for post-combustion carbon capture due to a large number of required GLMC modules and high head-losses of $V$ and $L$ streams.

Quek et al. [160] developed a predictive GLMC model for high-pressure NG sweetening. A temperature correction was calculated for the adiabatic GLMC to account for the ChA thermal effects. Model equations were implemented in gPROMS modeling language using a second-order centered finite difference scheme to discretize the differential equations. Model results attained good comparison with lab-scale experiments of $CO_2$ absorption from $CH_4$-$CO_2$ and $N_2$-$CO_2$ mixtures at P = 11 bar and from pilot-scale experiments on NG sweetening at P = 54 bar, both using 39 w/w% aqueous-MDEA with 5 w/w% PZ.

Quek et al. [161] indicated limitations of their work [160] since it fails to describe the gradual increase in temperature through the equipment, besides not considering solvent evaporation. Thus, an improvement of their GLMC model was carried out in [161] via combining 1D and 2D mass balance equations to include MPW. Real gas behavior is implemented via PR-EOS. The GLMC model is shown in Equations (20) to (26).

**Gas mass balance:**

$$\bar{v}_V(z)\frac{\partial \overline{C}_{V_{CO_2}}(z)}{\partial z} = -2\frac{\varepsilon D_{CO_2,V}}{\tau r_i}\left(1 - \frac{ZRT_V}{P_V}\right)\frac{\partial C_{V_{CO_2}}(r_i^+,z)}{\partial r} \tag{20}$$

$$\frac{\partial \bar{v}_V(z)}{\partial z} = -2\frac{\varepsilon D_{CO_2,V}}{\tau r_i}\left(\frac{ZRT_V}{P_V}\right)\frac{\partial C_{V_{CO_2}}(r_i^+,z)}{\partial r} \tag{21}$$

**Membrane-dry mass balance:**

$$0 = \frac{\partial^2 C_{CO_2}(r,z)}{\partial r^2} + \frac{1}{r}\frac{\partial C_{CO_2}(r,z)}{\partial r} \tag{22}$$

**Membrane-wet mass balance:**

$$0 = \frac{\varepsilon}{\tau}D_{k,L}\left[\frac{\partial^2 C_{L_k}(r,z)}{\partial r^2} + \frac{1}{r}\frac{\partial C_{L_k}(r,z)}{\partial r}\right] + R_k\left(C_{L_k}(r,z)\right), \ k \in \{CO_2, solvent\} \tag{23}$$

**Liquid mass balance:**

$$v_L(r)\frac{\partial C_{L_k}(r,z)}{\partial z} = D_{k,L}\left[\frac{\partial^2 C_{L_k}(r,z)}{\partial r^2} + \frac{1}{r}\frac{\partial C_{L_k}(r,z)}{\partial r} + \frac{\partial^2 C_{L_k}(r,z)}{\partial z^2}\right] + R_k\left(C_{L_k}(r,z)\right), \ k \in \{CO_2, solvent\} \tag{24}$$

$$v_L(r) = 2\bar{v}_L\left(r_{shell}^2 - r_o^2\right)\frac{r_{shell}^2 - r_o^2 + 2r_{shell}^2 \ln\left(\frac{r_o}{r_{shell}}\right)}{4r_o^2 r_{shell}^2 - r_o^4 - 3r_{shell}^4 - 4r_{shell}^4 \ln\left(\frac{r_o}{r_{shell}}\right)} \tag{25}$$

$$\bar{v}_L = \frac{F_L}{N_{HF}\Pi r_{shell}^2(1-\phi)} \tag{26}$$

where $\bar{v}_V$ and $\bar{v}_L$ are $V$ and $L$ average velocities, $\overline{C}_{V_{CO_2}}$ is the average $V$ concentration of $CO_2$, $D_{CO_2,V}$ is $V$ $CO_2$ diffusivity, $D_{k,L}$ is $L$ diffusivity of "$k$," $r_{shell}$ is the outer radius of the virtual cylindrical domain around each HFM, $r_1^+$ is the gas-membrane interface at the

pore-side, $F_L$ is $L$ volumetric flowrate, $N_{HF}$ is the number of HFM within GLMC module and $\varphi$ is the fiber volume fraction (packing-ratio).

To predict solvent evaporation in counter-current GLMC [161], the authors assumed a water-saturated gas at the GLMC outlet and used Raoult's law to calculate $H_2O$ content. However, there is some impertinency in this assumption, as the two premises for Raoult's law are: (i) ideal gas vapor phase at low-pressure; (ii) ideal solution liquid phase with chemically similar species. As the GLMC has a gas phase at high pressure in VLE with a non-ideal solution with high ionic interaction between species, it would be much more adequate to use an equation-of-state for vapor fugacities, besides an excess Gibbs free energy local composition model for aqueous electrolyte systems. At the end of the article, the authors admitted that Raoult's law is questionable. Moreover, an energy balance was performed [161], considering the heat losses of solvent evaporation. As there is an inaccuracy in solvent loss prediction, this step is probably also inaccurate. Moreover, Henry's law was employed to calculate liquid phase methane, ethane, and propane concentrations. These steps are shown in Equations (27) to (31).

**V composition:**

$$Y_k^{out} = X_k^{in} \frac{P_k^{sat}}{P_V}, \ k \in \{H_2O, solvent\} \tag{27}$$

**Rate of solvent loss:**

$$q_{V_k} = Y_k^{out} M_{V_k} \overline{v}_V(0) N_{HF} \Pi r_1^2 \frac{P_V}{ZRT_V} = X_k^{out} M_{V_k} \overline{v}_V(0) N_{HF} \Pi r_1^2 \frac{P_k^{sat}}{ZRT_V}, \ k \in \{H_2O, solvent\} \tag{28}$$

**Energy balance:**

$$F_L \rho_L C_{P_L} \frac{dT_L(z)}{dz} = N_{HF} \Pi r_1^2 \frac{\partial\left(\overline{v}_V(z) \overline{C}_{V_{CO_2}}(z)\right)}{\partial z} \Delta H_r \tag{29}$$

**$L$ light hydrocarbons concentrations:**

$$C_{L_k}^{out} = \frac{Y_k^{in} P_V}{H_k}, \ k \in \{CH_4, C_2H_6, C_3H_8\} \tag{30}$$

**Light hydrocarbons loss rate:**

$$q_{L_k}^{out} = C_{L_k}^{out} M_{L_k} F_L, \ k \in \{CH_4, C_2H_6, C_3H_8\} \tag{31}$$

where $Y_k^{out}$ is $V$ outlet mol fraction of "$k$," $X_k^{out}$ and $X_k^{in}$ are $L$ outlet/inlet mol fraction of "$k$," $P_k^{sat}$ is the saturation pressure of "$k$," $q_{V_k}$ is $V$ mass flowrate of "$k$," $M_{V_k}$ is $V$ molar mass of "$k$," $\overline{v}_V$ is $V$ average velocity and $C_{P_L}$ is $L$ mass isobaric heat capacity.

Besides ignoring solvent head-loss [161], the authors tried to convince us that gas head-loss can be safely neglected. On the other hand, several researchers, including Hoff and Svendsen [157], proved exactly the opposite. The model was solved with gPROMS modeling language. Gas feed compositions were 20–24 mol% $CO_2$ in a $N_2$-$CO_2$ mixture (P = 54 bar, T = 293 K) in lab experiments and 5–18 mol% $CO_2$ NG (P = 54 bar, T = 298 K) in pilot runs. Solvent was 37 w/w% aqueous-MDEA with 6 w/w% PZ (P = 54.30 bar, T = 310–313 K) in lab experiments and 39 w/w% aqueous-MDEA with 5 w/w% PZ (T = 303–308 K) in pilot runs.

Quek et al. [162] installed the GLMC model from [161] in the process simulator gPROMS to perform the techno-economic assessment of the complete process for high-pressure NG sweetening and low-pressure regeneration in a closed-loop. Simulations predicted $CO_2$ absorption and light hydrocarbons losses, solvent evaporation, besides process energy consumption. Results were compared with data from a pilot plant in Malaysia with reasonable agreement. Raw gas composition was $CO_2$ = 5 mol%, $CH_4$ = 85 mol%, $C_2H_6$ = 7 mol%, $C_3H_8$ = 3 mol% at P = 54 bar, T = 298 K, and the fresh solvent was

39 w/w% aqueous-MDEA with 5 w/w% PZ (P = 54.3 bar, T = 303–308 K). Two solvent regenerations were considered: regeneration in stripping columns or semi-regeneration in low-pressure flash separators. Posteriorly, the design and scale-up of a commercial-scale GLMC was conducted with a semi-lean solvent at raw gas composition $CO_2$ = 24 mol%, $CH_4$ = 73 mol%, $C_2H_6$ = 2 mol%, $C_3H_8$ = 1 mol% (P = 54 bar, T = 298 K), while the semi-lean solvent was 39 w/w% aqueous-MDEA with 5 w/w% PZ (P = 54.3 bar, T = 318 K). $CO_2$ content was abated from 24 mol% to 6.5 mol%, which the authors considered excellent as they recommended the semi-lean regeneration with low-pressure flash to reduce energy consumption. However, it is widely known that the $CO_2$ target for NG sweetening usually ranges from 2 to 3 mol% [14]; thereby, employing stripping columns for complete solvent regeneration is more adequate in this case.

Da Cunha et al. [128] performed a technical evaluation of GLMC $CO_2$ removal from $CO_2$-rich NG using 30 w/w% aqueous-MEA through process simulations in Aspen HYSYS V8.8. The authors developed a GLMC model, with the driving force for $CO_2$ transfer predicted with $CO_2$ $V$ and $L$ fugacity differences, rigorously calculated with thermodynamic resources of Aspen HYSYS V8.8 provided by the Acid-Gas package [163] (based on Electrolyte-NRTL model) for liquid phase properties and PR-EOS for vapor properties [164]. GLMC model equations consist of a non-linear algebraic system in Equations (32) to (34) solved by Newton-Raphson Method.

$$N_k = \Pi_k A \left\{ \frac{\left( \hat{f}_{V_{CO_2}}^{in} - \hat{f}_{L_{CO_2}}^{out} \right) - \left( \hat{f}_{V_{CO_2}}^{out} - \hat{f}_{L_{CO_2}}^{in} \right)}{\ln\left( \hat{f}_{V_{CO_2}}^{in} - \hat{f}_{L_{CO_2}}^{out} \right) - \ln\left( \hat{f}_{V_{CO_2}}^{out} - \hat{f}_{L_{CO_2}}^{in} \right)} \right\} \tag{32}$$

$$N_k = L^{out} X_{CO_2}^{out} - L^{in} X_{CO_2}^{in} \tag{33}$$

$$N_k = V^{in} Y_{CO_2}^{in} - V^{out} Y_{CO_2}^{out} \tag{34}$$

where $N_k$ is the trans-membrane transfer rate of component "$k$," $\Pi_k$ is the trans-membrane mass transfer coefficient of "$k$," $A$ is the transfer area of the GLMC battery, $\hat{f}_{V_{CO_2}}$ and $\hat{f}_{L_{CO_2}}$ are $V$ and $L$ $CO_2$ fugacities, $Y_{CO_2}$ and $X_{CO_2}$ are $V$ and $L$ $CO_2$ molar fractions, and $V$ and $L$ are molar flowrates.

To perform process simulations in HYSYS, the authors created a GLMC HYSYS Unit Operation Extension (GLMC-UOE), in a similar procedure to Arinelli et al. [165] for MP unit operation. GLMC-UOE was programmed in Visual Basic 6.0, which contains an HYSYS Type Library with commands to access HYSYS directly. After programming and compiling the code, an external DLL file was generated, which was installed through an EDF file as an available unit operation in the HYSYS palette usable in HYSYS flowsheets. The EDF file was programmed with the Aspen HYSYS Extension View Editor and also provided the GLMC-UOE property windows, where an HYSYS user interface is filled by the user with all the necessary simulation information. GLMC model equations were programmed to be solved by the Newton–Raphson Method.

GLMC-UOE [128] is worthy of note because its present version already represents a new unit operation GLMC, which is implemented within an up-to-date professional simulation environment using full thermodynamics to calculate fluid properties and fugacities of real liquid and gas phases. Moreover, GLMC-UOE can be integrated into processing flowsheets comprising several other unit operations offered in the HYSYS palette, eventually constituting very complex process flowsheets.

### 4.2. GLMC Modeling for $CO_2$ Stripping from the Solvent

Complete GLMC models for low-pressure $CO_2$ stripping from the solvent require the same framework of GLMC for $CO_2$ removal from NG, as described in the first paragraph of Section 4.1.

Some GLMC models were developed for $CO_2$ stripping from physical solvents. Mehdipourghazi et al. [166] investigated $CO_2$ stripping from the water via PVDF-HFM

GLMC. The 2D model (Table 7, Equations (48), (50)–(53), (55), and (56)) only considers isothermal $CO_2$ transfer. Lee et al. [167] also studied vacuum-driven GLMC $CO_2$ stripping from water through experiments and using a 1D GLMC model (Table 6, Equations (35)–(38)) to assess liquid flowrate, vacuum pressure, and pH. Bahlake et al. [168] investigated GLMC $CO_2/H_2S$ simultaneous stripping from the water via a 3D modeling approach whose differential equations were solved with COMSOL software. Sohaib et al. [69] investigated two ionic liquids for post-combustion carbon capture via GLMC-absorber coupled to GLMC-stripper. A 2D dynamic model (Table 7, Equations (48), (50)–(53), (55), and (56))was implemented based on low-pressure and low-temperature mass transfer equations. Vadillo et al. [169] developed a 1D-2D GLMC model (Table 7, Equation (55), Table 8, Equation (68)) for vacuum-driven regeneration of the ionic–liquid 1-ethyl-3-methylimidazolium acetate [emin][Ac] solvent considering isothermal conditions. The best operating conditions were achieved at P = 0.4 bar and T = 310 K.

The literature has an evident interest in GLMC $CO_2$ stripping from aqueous-MEA. Ghadiri et al. [170] used pure $N_2$ for $CO_2$ stripping in a GLMC modeling study. The 2D model (Table 7, Equations (48), (53) and (55)–(57)) assumed isothermal and steady-state conditions, only $CO_2$ transference, and VLE via Henry's law. $CO_2$ mass transfer predictions were compared with experimental data from [113], with reasonable agreement. To remove $CO_2$ from 20 w/w% aqueous-MEA at T = 353 K, Wang et al. [171] used vacuum-driven stripping with steam as a stripping agent on the shell side at $P$ = 20 kPa. The 1D-2D model (Table 7, Equation (55), Table 8, Equation (68)) is based on a system of steady-state continuity equations for each species with simultaneous diffusion and chemical reaction, considering isothermal operation and VLE via Henry's law. Mohammed et al. [172] developed a 2D GLMC model (Table 7, Equations (48), (50)–(53), (55), and (56)) to investigate the effect of $SiO_2$ nanoparticles on $CO_2$ stripping from aqueous-MEA. Authors considered isothermal GLMC and VLE via Henry's law. No energy balances or head-loss calculations were performed.

**Table 6.** 1D GLMC models are not related to processing simulators.

| Main Equations | | References |
|---|---|---|
| V→L (positive direction by convention) | | |
| **Mass balance equations:** | | |
| $\frac{dV_k}{dz} = -N_{HF}\Pi d_o N_k^{flux}$ | (35) | [167,173,174] |
| $\frac{dL_k}{dz} = N_{HF}\Pi d_o N_k^{flux}$ | (36) | [167,173,174] |
| $N_k^{flux} = k_{k,ov}\left(C_{V_k} - C_{L_k}^*\right)$ | (37) | [135–137,167,173,174] |
| $\frac{1}{a_m k_{k,ov}} = \frac{1}{a_{ext}k_{k,V}} + \frac{1}{a_m k_{k,m}} + \frac{H_k}{a_{int}Ek_{k,L}}$ | (38) | [167,174] |
| **Energy balance equations:** | | |
| $\frac{dT_V}{dz} = -\frac{Q^{flux}}{\overline{C}_{P_V}\sum_k V_k^{flux}}$ | (39) | [173,174] |
| $\frac{dT_L}{dz} = \frac{Q^{flux} + \sum_k N_k^{flux}\Delta H_{k,abs/evap} + N_{CO_2}^{flux}\Delta H_r}{\overline{C}_{P_L}\sum_k L_k^{flux}}$ | (40) | [173] |
| $\frac{dT_L}{dz} = \frac{Q^{flux} + \sum_k N_k^{flux}\Delta H_{k,abs/evap}}{\overline{C}_{P_L}\sum_k L_k^{flux}}$ | (41) | [174] |
| $Q^{flux} = U(T_V - T_L)$ | (42) | [173,174] |
| $\frac{1}{a_m U} = \frac{1}{a_{ext}\Omega_V} + \frac{1}{a_m\Omega_m} + \frac{1}{a_{int}\Omega_L}$ | (43) | [174] |
| **Pressure drop calculation:** | | |
| Stream inside HFM (Hagen-Poiseuille equation): | | |
| $\frac{dP_i}{dz} = -\frac{8\mu_i v_i}{(r_i)^2}$ | (44) | [173,174] |
| Stream in the shell-side (Happel equation): | | |
| $\frac{dP_o}{dz} = -\mu_o v_o K_{koz}$ | (45) | [173,174] |
| $K_{koz} = \frac{8\varphi}{\left[2ln\left(\frac{1}{\varphi}\right)-3+4\varphi-\varphi^2\right](r_o)^2}$ | (46) | [173,174] |

where: $k_{k,ov}$: overall mass transfer coefficient of "$k$" (m/s); $Q^{flux}$: heat flux (W/m²); $V_k^{flux}$: V molar flux of "$k$" (mol/s); $\Delta H_{k,abs/evap}$: enthalpy of absorption or vaporization (J/mol); $L_k^{flux}$: L molar flux of "$k$" (mol/s); U: overall heat transfer coefficient (W/(m².K)); $\Omega_L$: L heat transfer coefficient (W/(m².K)); $a_{ext}$: external specific interfacial area (-); $a_{int}$: internal specific interfacial area (-); $a_m$: specific membrane area (-); $P_i$: Pressure of the stream inside HFM (bar); $\mu_i$: Dynamic viscosity of the predominant phase inside HFM (Pa.s); $v_i$: Velocity of the stream at internal surface of HFM (m/s); $P_o$: Pressure in the shell-side (bar); $\mu_o$: Dynamic viscosity of the predominant phase in the shell-side (Pa.s); $v_o$: Velocity of the stream at the external surface of HFM (m/s); $K_{koz}$: Kozeny factor (m⁻²).

**Table 7.** GLMC 2D models are not related to processing simulators.

| Main Equations | | References |
|---|---|---|
| **Mass balance equations:** | | |
| Reaction rate: | | |
| $R_k = k_r C_{L_k} C_{L_j}$ | (47) | [60,65,76,79,80,84,90,130–134,139,143–145,149,150] |
| Shell-side with Happel free surface model for velocity distribution: | | |
| $D_{k,o}\left[\frac{\partial^2 C_{k,o}}{\partial r^2} + \frac{1}{r}\frac{\partial C_{k,o}}{\partial r} + \frac{\partial^2 C_{k,o}}{\partial z^2}\right] + R_k = v_{z,o}\frac{\partial C_{k,o}}{\partial z}$ | (48) [@] | [65,69,76,79,80,82,84,90,130–134,139,141,143–145,147,148,150,166,170,172] |
| $D_{k,o}\left[\frac{\partial^2 C_{k,o}}{\partial r^2} + \frac{1}{r}\frac{\partial C_{k,o}}{\partial r} + \frac{\partial^2 C_{k,o}}{\partial z^2}\right] + R_k = \frac{\partial}{\partial z}\left(v_{z,o} C_{k,o}\right) + \frac{1}{r}\frac{\partial}{\partial r}\left(r v_{r,o} C_{k,o}\right)$ | (49) [@] | [60] |
| $v_{z,o} = 2\bar{v}_{z,o}\left[1 - \left(\frac{r}{r_i}\right)^2\right] \times \left[\frac{\left(\frac{r}{r_{shell}}\right)^2 - \left(\frac{r_o}{r_{shell}}\right)^2 + 2ln\left(\frac{r_o}{r}\right)}{3 + \left(\frac{r_o}{r_{shell}}\right)^4 - 4\left(\frac{r_o}{r_{shell}}\right)^2 + 4ln\left(\frac{r_o}{r_{shell}}\right)}\right]$ | (50) | [60,65,69,76,79,80,84,90,130–134,141,143–145,147,148,150,166,172] |
| $r_{shell} = r_o\sqrt{\frac{1}{\varphi}}$ | (51) | [60,65,69,76,79,80,82,84,90,130,132–134,139,141,143–145,147,148,150,166,172] |
| $\varphi = 1 - N_{HF}\left(\frac{d_o}{D}\right)^2$ | (52) | [69,79,80,82,90,145,166,172] |
| Inside HFM with velocity distribution by Newtonian laminar flow: | | |
| $D_{k,i}\left[\frac{\partial^2 C_{k,i}}{\partial r^2} + \frac{1}{r}\frac{\partial C_{k,i}}{\partial r} + \frac{\partial^2 C_{k,i}}{\partial z^2}\right] + R_k = v_{z,i}\frac{\partial C_{k,i}}{\partial z}$ | (53) [#] | [65,69,76,79,80,82,84,90,130,132–134,139,141,143–145,147,148,150,166,170,172] |
| $D_{k,i}\left[\frac{\partial^2 C_{k,i}}{\partial r^2} + \frac{1}{r}\frac{\partial C_{k,i}}{\partial r} + \frac{\partial^2 C_{k,i}}{\partial z^2}\right] + R_k = \frac{\partial}{\partial z}\left(v_{z,i} C_{k,i}\right) + \frac{1}{r}\frac{\partial}{\partial r}\left(r v_{r,i} C_{k,i}\right)$ | (54) [#] | [60] |
| $v_{z,i} = 2\bar{v}_{z,i}\left[1 - \left(\frac{r}{r_i}\right)^2\right]$ | (55) | [65,69,76,79,80,82,84,90,130–134,141,143–145,147,148,150,166,170,172] |
| Diffusion through gas-filled membrane pores: | | |
| $D_{k,m}\left[\frac{\partial^2 C_{k,m}}{\partial r^2} + \frac{1}{r}\frac{\partial C_{k,m}}{\partial r} + \frac{\partial^2 C_{k,m}}{\partial z^2}\right] = 0$ | (56) | [60,65,69,76,79,80,82,84,90,130,132–134,139,141,143–145,147,148,150,166,170,172] |
| $D_{k,m} = \frac{D_{k,i} \times \varepsilon}{\tau}$ or $D_{k,m} = \frac{D_{k,o} \times \varepsilon}{\tau}$ | (57) [$] | [76,80,84,90,132,134,141,143–145,147,170] |
| Diffusion through partially-wetted membrane pores: | | |
| $Gas - filled\ part: D_{k,mV}\left[\frac{\partial^2 C_{k,mV}}{\partial r^2} + \frac{1}{r}\frac{\partial C_{k,mV}}{\partial r} + \frac{\partial^2 C_{k,mV}}{\partial z^2}\right] = 0$ | (58) | [76,80,144] |
| $Liquid - filled\ part: D_{k,mL}\left[\frac{\partial^2 C_{k,mL}}{\partial r^2} + \frac{1}{r}\frac{\partial C_{k,mL}}{\partial r} + \frac{\partial^2 C_{k,mL}}{\partial z^2}\right] + R_{k,mL} = 0$ | (59) | [76,80,144] |
| $R_{k,mL} = R_k \times \varepsilon$ | (60) [%] | [76,80,144] |
| **Momentum balance equations:** | | |
| Shell-side: | | |
| $Radial - direction: \rho_o\left[v_{r,o}\frac{\partial v_{r,o}}{\partial r} + v_{z,o}\frac{\partial v_{r,o}}{\partial z}\right] = -\frac{\partial P_o}{\partial r} + \mu_o\left\{\frac{\partial}{\partial r}\left[\frac{1}{r}\frac{\partial}{\partial r}\left(r v_{r,o}\right) + \frac{\partial^2 v_{r,o}}{\partial z^2}\right]\right\}$ | (61) | [60,84] |

**Table 7.** *Cont.*

| Main Equations | | References |
|---|---|---|
| Axial − direction : $\rho_o\left[v_{r,o}\frac{\partial v_{z,o}}{\partial r} + v_{z,o}\frac{\partial v_{z,o}}{\partial z}\right] = -\frac{\partial P_o}{\partial z} + \mu_o\left\{\frac{\partial}{\partial r}\left[\frac{1}{r}\frac{\partial}{\partial r}\left(rv_{z,o}\right) + \frac{\partial^2 v_{z,o}}{\partial z^2}\right]\right\}$ | (62) | [60,84] |
| Inside HFM: | | |
| Radial − direction : $\rho_i\left[v_{r,i}\frac{\partial v_{r,i}}{\partial r} + v_{z,i}\frac{\partial v_{r,i}}{\partial z}\right] = -\frac{\partial P_i}{\partial r} + \mu_i\left\{\frac{\partial}{\partial r}\left[\frac{1}{r}\frac{\partial}{\partial r}\left(rv_{r,i}\right) + \frac{\partial^2 v_{r,i}}{\partial z^2}\right]\right\}$ | (63) | [60,84] |
| Axial − direction : $\rho_i\left[v_{r,i}\frac{\partial v_{z,i}}{\partial r} + v_{z,i}\frac{\partial v_{z,i}}{\partial z}\right] = -\frac{\partial P_i}{\partial z} + \mu_i\left\{\frac{\partial}{\partial r}\left[\frac{1}{r}\frac{\partial}{\partial r}\left(rv_{z,i}\right) + \frac{\partial^2 v_{z,i}}{\partial z^2}\right]\right\}$ | (64) | [60,84] |
| Energy balance equations: | | |
| Shell-side: | | |
| $\Omega_o\left[\frac{\partial^2 T_o}{\partial r^2} + \frac{1}{r}\frac{\partial T_o}{\partial r} + \frac{\partial^2 T_o}{\partial z^2}\right] + R_k\Delta H_r = \rho_o C_{P_o}\left(v_{r,o}\frac{\partial T_o}{\partial r} + v_{z,o}\frac{\partial T_o}{\partial z}\right)$ | (65) [@] | [60] |
| Inside HFM: | | |
| $\Omega_i\left[\frac{\partial^2 T_i}{\partial r^2} + \frac{1}{r}\frac{\partial T_i}{\partial r} + \frac{\partial^2 T_i}{\partial z^2}\right] + R_k\Delta H_r = \rho_i C_{P_i}\left(v_{r,i}\frac{\partial T_i}{\partial r} + v_{z,i}\frac{\partial T_i}{\partial z}\right)$ | (66) [#] | [60] |
| HFM: | | |
| $\Omega_m\left[\frac{\partial^2 T_m}{\partial r^2} + \frac{1}{r}\frac{\partial T_m}{\partial r} + \frac{\partial^2 T_m}{\partial z^2}\right] = 0$ | (67) | [60] |

where: $k_r$: kinetic constant for "$k$" (m/s); $C_{L_j}$: L molar concentration of "$j$" (mol/m$^3$); $D_{k,o}$: diffusivity of "$k$" in the shell-side (mol/m$^3$); $C_{k,o}$: molar concentration of "$k$" in the shell-side (mol/m$^3$); $v_{z,o}$: velocity in axial direction in the shell-side (m/s); $v_{r,o}$: velocity in radial direction in the shell-side (m/s); $\bar{v}_{z,o}$: average velocity in axial direction in the shell-side (m/s); $d_o$: external HFM diameter (m); $D_{k,i}$: diffusivity of "$k$" inside HFM (mol/m$^3$); $C_{k,i}$: molar concentration of "$k$" inside HFM (mol/m$^3$); $v_{z,i}$: velocity in axial direction inside HFM (m/s); $v_{r,i}$: Velocity in radial direction inside HFM (m/s); $\bar{v}_{z,i}$: average velocity in axial direction inside HFM (m/s); $D_{k,m}$: Diffusivity of "$k$" inside membrane pores (mol/m$^3$); $C_{k,m}$: molar concentration of "$k$" inside membrane pores (mol/m$^3$); $D_{k,\mathrm{mV}}$: diffusivity of "$k$" inside the gas-filled part of HFM pores (mol/m$^3$); $C_{k,\mathrm{mV}}$: molar concentration of "$k$" inside the gas-filled part of HFM pores (mol/m$^3$); $D_{k,\mathrm{mL}}$: diffusivity of "$k$" inside the liquid-filled part of HFM pores (mol/m$^3$); $C_{k,\mathrm{mL}}$: molar concentration of "$k$" inside the liquid-filled part of HFM pores (mol/m$^3$); $R_{k,mL}$: chemical reaction rate of "$k$" inside the liquid-filled part of HFM pores (mol/(m$^3$.s)); $\rho_o$: density in the shell-side (kg/m$^3$); $\rho_i$: density inside HFM (kg/m$^3$); $\Omega_o$: heat transfer coefficient of stream in the shell-side (W/(m$^2$.K)); $T_o$: shell-side temperature (K); $C_{P_o}$: mass isobaric heat capacities in the shell-side (J/(kg.K)); $\Omega_i$: heat transfer coefficient inside HFM (W/(m$^2$.K)); $T_i$: HFM inside temperature (K); $C_{P_i}$: mass isobaric heat capacities inside HFM (J/(kg.K)); $T_m$: HFM temperature (K). [@] $R_k$ term only appears in this equation if the solvent flows in the shell-side; [#] $R_k$ term only appears in this equation if the solvent flows inside the HFM; [$] The diffusivity-term of the equation depends on which side of HFM the gas flows; [%] In membrane section, chemical reactions occur just inside the pores.

**Table 8.** 1D-2D models not related to processing simulators.

| Main Equations | | References |
|---|---|---|
| **Mass balance equation:** | | |
| $D_{k,i}\left[\frac{\partial^2 C_{k,i}}{\partial r^2} + \frac{1}{r}\frac{\partial C_{k,i}}{\partial r}\right] - R_{k,i} = v_{z,i}\frac{\partial C_{k,i}}{\partial z}$ | (68) | [78,169,171] |
| Velocity distribution: (Newtonian laminar flow) Equation (55) | | [78,169,171] |

The following works were conducted in the context of post-combustion carbon capture. They are considered in the present review as low-pressure GLMC $CO_2$ stripping from the solvent is conducted similarly for both GLMC processes for carbon capture from NG and flue gas. Scholes [173] used a simple model and a 1D rigorous model (Table 6, Equations (35)–(37), (39), (40), (42), and (44)–(46)) with mass/energy balances and pressure drop calculation to study vacuum-driven $CO_2$ stripping from aqueous-MEA 30 w/w%, using steam as stripping-gas. The best stripping temperature was found between 90 °C and 105 °C. Above this temperature range, the solvent starts vaporizing, and below this range, the mass transfer driving force is drastically reduced. Scholes highlighted the trade-off between the reduction of stripping heat requirement against the vacuum pump power requirement. Zaidiza et al. [174] developed a GLMC model for post-combustion $CO_2$ absorption with aqueous-MEA solvent and for the respective $CO_2$ stripping. The 1D adiabatic model (Table 6, Equations (35)–(39) and (41)–(46)) considers the bidirectional mass transfer of all species ($CO_2$, $H_2O$, MEA); energy balances; and head-loss calculations via Hagen-Poiseuille equation for the HFM stream and via Happel equation [175] for the shell-side. Model equations were presented with dimensional problems and inverted signs (e.g., Pa.s$^{-1}$ was the viscosity unit instead of Pa.s). Model assumptions include steady-state, ideal gas behavior, and VLE via Henry's law. Water condensation/evaporation was considered at the liquid-membrane interface. The authors emphasize that GLMC-stripper needs HFM resistant to high temperatures (≈120 °C). Also, since water can be transferred in both directions, capillary condensation can occur, which would negatively influence mass transfer performance similarly to MPW.

### 4.3. GLMC Models Not Related to Process Simulators

The main GLMC models not related to processing simulators are presented in Table 6 for 1D models, Table 7 for 2D models, and Table 8 for 1D-2D models. These models were already discussed in Section 4.1 and are detailed here for completeness. They usually lack thermodynamic frameworks and present several simplifications, such as isothermal GLMC and ideal gas behavior. These tables list equations and modeling strategies adopted, showing that several researchers used the same equations. Table 9 presents the used software for the numerical solution of GLMC equations.

**Table 9.** Software for solving GLMC models.

| Software | References |
|---|---|
| MATLAB | [33,69,78,135–137,171,173,174] |
| OpenFOAM | [146] |
| COMSOL | [60,65,69,76,79,82,84,90,130–134,139,141,143–145,147–150,168,170,172] |
| PARDISO | [166] |
| ANSYS | [142] |

## 5. Concluding Remarks and Future Prospects

Aqueous-amine solvents are the most studied in high-pressure NG sweetening and in solvent regeneration with GLMC. Regarding physical solvents, the increasing interest in ionic liquids is highlighted.

Both experimental and modeling GLMC works on carbon capture from $CO_2$-rich NG must consider all the real offshore operating conditions: (i) high-pressure operation;

(ii) mass transfer of all species, including $CO_2$, $H_2S$, $CH_4$, $H_2O$, and other light hydrocarbons; (iii) thermal effects, with outlet temperatures assessment; (iv) head-losses of both phases.

Several experimental works were restricted to $CO_2$ mass transfer at low pressures. The most useful experimental GLMC works so far investigated high-pressure simultaneous $CO_2/H_2S$ removal from gas mixtures containing $CH_4$, as in [100,101]. Experimental works were usually focused on inlet conditions optimizations to enhance $CO_2$ mass transfer. To our knowledge, there are no works in the literature providing data on water mass transfer. Pressure drop is frequently neglected in the bench scale but should be investigated at least in the pilot scale. Data of light hydrocarbons ($C_2^+$) mass transfer are also scarce. Therefore, future experimental works should perform complete outlet conditions assessments, providing a complete understanding of GLMC behavior.

The development of GLMC models not related to processing simulators is a complex work and imposes an additional task: the supply of a thermodynamic framework for correct RVLE calculation and thermodynamic and transport properties prediction. This was done in de Medeiros et al. [33], in which a full thermodynamic framework was provided for simulation of $CO_2$ removal from high-pressure NG with a 1D GLMC model considering rigorous multicomponent mass, energy, and momentum balances.

Among the scarce GLMC models that apply at least mass, energy, and momentum balances, Ghasem et al. [60] provided a 2D model for $CO_2$ absorption from NG, while Scholes [173] and Zaidiza et al. [174] provided 1D models for $CO_2$ stripping from the solvent. On the other hand, it can be seen in Tables 6–9 that several 2D models are performing $CO_2$ mass transfer with simplifications such as isothermal condition, ideal gas behavior, and zero pressure drop, using COMSOL or CFD to solve GLMC equations via finite elements. It can be concluded that these models are simplified and cannot reproduce industrial GLMC behavior for NG conditioning with accuracy. Typical examples are [80,150], which proposed 2D models for isothermal $CO_2$ absorption from NG with aqueous-amine solvents.

GLMC modeling and its implementation in up-to-date process simulators are of great importance to enable industrial-scale technical-economic assessment. In [156,157], mass, energy, and momentum balances were considered, but the multicomponent mass transfer was not. In [161,162], despite considering energy balance and multicomponent mass transfer, the thermodynamic framework is inadequate, and there is no pressure drop calculation. Da Cunha et al. [128] developed a GLMC-UOE to access HYSYS full thermodynamics. Just $CO_2$ mass transfer was considered, and the model lacks energy and momentum balances. The point here is that to obtain a complete GLMC model to be installed in process simulators, the model must gather all the positive aspects of [128,156,157,161,162].

Moreover, it is worthwhile to think of distributed parameters models as in [33] for GLMC-absorber and [174] for GLMC-stripper. The work of Lock et al. [176] created an MP unit operation extension (MP-UOE) installed in Aspen HYSYS with the same method as da Cunha et al. [128] for GLMC-UOE. To construct the MP-UOE distributed model [176], the succession of states approach was employed, in which the equipment is divided into many finite cells in the axial direction. In the same way, it would be interesting to develop a GLMC complete distributed model to be installed in Aspen HYSYS as GLMC-UOE or in other available process simulators.

**Author Contributions:** Conceptualization J.L.d.M.; methodology, J.L.d.M. and O.d.Q.F.A.; formal analysis, G.P.d.C., J.L.d.M., and O.d.Q.F.A.; investigation, G.P.d.C. and J.L.d.M.; data curation, G.P.d.C.; writing—original draft preparation, G.P.d.C. and J.L.d.M.; visualization, G.P.d.C.; writing—review and editing, J.L.d.M. and O.d.Q.F.A.; supervision, J.L.d.M. and O.d.Q.F.A.; project administration, J.L.d.M. and O.d.Q.F.A. All authors have read and agreed to the published version of the manuscript.

**Funding:** J.L.d.M. and O.d.Q.F.A. acknowledge financial support from CNPq-Brazil (313861/2020-0 and 312328/2021-4).

**Institutional Review Board Statement:** Not applicable.

**Informed Consent Statement:** Not applicable.

**Data Availability Statement:** Not applicable.

**Conflicts of Interest:** The authors declare no conflict of interest.

## Abbreviations

| | |
|---|---|
| 1D | One-Dimensional |
| 2D | Two-Dimensional |
| AGWA | Acid-Gas Water Alkanolamines |
| CCS | Carbon Capture and Storage |
| CCUS | Carbon Capture Utilization and Storage |
| CFD | Computational Fluid Dynamics |
| ChA | Chemical-Absorption |
| DEPG | Dimethyl Ether of Polyethylene Glycol |
| DLL | Dynamic-Link Library |
| EDF | Extension Definition File |
| EMWR | Equivalent Moles Without Reaction |
| e-NRTL | Electrolyte Non-Random Two Liquid |
| EGR | Enhanced Gas Recovery |
| EOR | Enhanced Oil Recovery |
| GHG | Greenhouse Gas |
| GLMC | Gas-Liquid Membrane Contactor |
| GLMC-UOE | Gas-Liquid Membrane Contactor HYSYS Unit Operation Extension |
| HFM | Hollow-Fiber Membrane |
| MDEA | Methyldiethanolamine |
| MEA | Monoethanolamine |
| $MMNm^3/d$ | Millions of Normal Cubic Meters per Day |
| MP | Membrane Permeation |
| MP-UOE | MP-UOE Membrane Permeation Unit Operation Extension |
| MPW | Membrane Pore Wetting |
| NG | Natural Gas |
| NRM | Newton-Raphson Method |
| ODE | Ordinary Differential Equations |
| PC-SAFT | Perturbed Chain Statistical Association Fluid Theory Equation-of-State |
| PFD | Process Flow Diagram |
| PMP | Partial Molar Property |
| PP | Polypropylene |
| PPO | Polyphenylene Oxide |
| PR | Peng-Robinson Equation-of-State |
| PSA | Pressure Swing Adsorption |
| RVLE | Reactive Vapor-Liquid Equilibrium |
| UOE | Unit Operation Extension |
| VLE | Vapor-Liquid Equilibrium |
| WDPA | Water Dew-Point Adjustment |

## Nomenclature

| | |
|---|---|
| $A_{SISA}$ | *Specific inner surface area of GLMC module ($m^2/m^3$)* |
| $A_E$, $A_I$ | *External/internal convective heat exchange areas ($m^2$)* |
| $A_{CSHF}$ | *Cross-section area inside HFM ($m^2$)* |
| $A_{CSS}$ | *Cross-section area within GLMC shell and outside HFM ($m^2$)* |
| $a_{ext}$ | *External specific interfacial area (-)* |

| | |
|---|---|
| $a_{int}$ | *Internal specific interfacial area (-)* |
| $a_m$ | *Specific membrane area (-)* |
| $A$ | *Mass transfer area of the GLMC battery ($m^2$)* |
| $C_{V_k}$, $C_{L_k}$ | *Molar concentration of "k" in V stream and L stream (mol/$m^3$)* |
| $C_{k,i}$ | *Molar concentration of "k" inside HFM (mol/$m^3$)* |
| $C_{k,m}$ | *Molar concentration of "k" inside membrane pores (mol/$m^3$)* |
| $C_{k,mL}$ | *Molar concentration of "k" inside the liquid-filled part of HFM pores (mol/$m^3$)* |
| $C_{k,mV}$ | *Molar concentration of "k" inside the gas-filled part of HFM pores (mol/$m^3$)* |
| $C_{k,o}$ | *Molar concentration of "k" in the shell side (mol/$m^3$)* |
| $C_{L_j}$ | *Molar concentration of "j" in L (mol/$m^3$)* |
| $C_{L_k}^*$ | *Molar concentration of "k" in L at equilibrium (mol/$m^3$)* |
| $C_{L_k,m}$ | *Concentration of "k" at liquid-membrane interface (mol/$m^3$)* |
| $\overline{C}_{V_{CO_2}}$ | *Average V concentration of $CO_2$ (mol/$m^3$)* |
| $\overline{C}_{P_V}$, $\overline{C}_{P_L}$ | *V and L molar isobaric heat capacities (kJ/(kmol.K))* |
| $\overline{C}_{P_{V,k}}$ | *V partial molar isobaric heat capacities of species "k" (kJ/(kmol.K))* |
| $C_{P_i}$ | *Mass isobaric heat capacities of stream inside HFM (J/(kg.K))* |
| $C_{P_o}$ | *Mass isobaric heat capacities in the shell side (J/(kg.K))* |
| $C_{P_V}$, $C_{P_L}$ | *V and L mass isobaric heat capacities (J/(kg.K))* |
| $D$ | *GLMC shell internal diameter (m)* |
| $D_{CO_2,V}$ | *V $CO_2$ diffusivity ($m^2$/s)* |
| $D_{k,m}^{eff}$ | *Effective membrane diffusivity of "k" ($m^2$/s)* |
| $D_k$ | *Molecular diffusivity of "k" ($m^2$/s)* |
| $D_{k,L}$ | *L diffusivity of "k" ($m^2$/s)* |
| $D_{Kn}$ | *Knudsen diffusivity ($m^2$/s)* |
| $D_m$ | *Membrane diffusivity ($m^2$/s)* |
| $D_{m,L}$ | *Diffusivity of membrane at liquid phase ($m^2$/s)* |
| $D_p$ | *Combined diffusivity ($m^2$/s)* |
| $D_{k,i}$ | *Diffusivity of "k" inside HFM (mol/$m^3$)* |
| $D_{k,m}$ | *Diffusivity of "k" inside membrane pores (mol/$m^3$)* |
| $D_{k,mL}$ | *Diffusivity of "k" inside the liquid-filled part of HFM pores (mol/$m^3$)* |
| $D_{k,mV}$ | *Diffusivity of "k" inside the gas-filled part of HFM pores (mol/$m^3$)* |
| $D_{k,o}$ | *Diffusivity of "k" in the shell-side (mol/$m^3$)* |
| $d_i$, $d_o$ | *Internal/external HFM diameter (m)* |
| $E$ | *Enhancement factor* |
| $E_{V_k}$, $E_{L_k}^{EMWR}$ | *PMP$^{EMWR}$ energies of "k" in V and L (kJ/mol)* |
| $F_L$ | *L volumetric flowrate ($m^3$/s)* |
| $\hat{f}_{L_k}^{in}$, $\hat{f}_{L_k}^{out}$ | *Vector of L inlet/outlet fugacities of "k" (bar)* |
| $\hat{f}_{V_K}$, $\hat{f}_{L_K}$ | *V and L fugacities of real species "k" (bar)* |
| $g$ | *Gravity acceleration (9.81 m/$s^2$)* |
| $\Delta H_{k,abs/evap}$ | *Enthalpy of absorption or vaporization (kJ/kmol);* |
| $\Delta H_r$ | *Enthalpy of reaction (kJ/kmol)* |
| $k_{k,V}$ | *Gas film mass transfer coefficient of "k" (mol/($m^2$.s.kPa))* |
| $k_{k,L}$ | *Liquid film mass transfer coefficient of "k" (mol/($m^2$.s.kPa))* |
| $k_{k,m}$ | *Membrane mass transfer coefficient of "k" (mol/($m^2$.s.kPa))* |
| $k_{k,ov}$ | *Overall mass transfer coefficient of "k" (m/s)* |
| $K_{koz}$ | *Kozeny factor ($m^{-2}$)* |
| $k_r$ | *Kinetic constant for "k" (m/s)* |
| $L_k^{flux}$ | *L stream molar flux of "k" (mol/s)* |
| $L_k$ | *L molar flowrate of "k" (mol/s)* |
| $M_V$, $M_L$ | *V and L EMWR molar masses (kg/mol)* |
| $M_{V_k}$, $M_{L_k}$ | *V and L molar masses of "k" (kg/mol)* |
| $N_{HF}$ | *Number of HFM within GLMC module* |
| $N_k^{flux}$ | *Trans-membrane fluxes of real species "k" (mol/(s.$m^2$))* |
| $N_k$ | *Trans-membrane mass transfer rates of "k" (mol/s)* |

| | |
|---|---|
| $N_M$ | Number of GLMC modules |
| $n_{RS}$ | Number of real species |
| $P_k^{sat}$ | Saturation pressure of "k" (bar) |
| $P_i$ | Pressure inside HFM (bar) |
| $P_o$ | Pressure in the shell-side (bar) |
| $P_V, P_L$ | V and L absolute pressures (bar) |
| $P_{V_k}$ | Partial pressure of "k" (bar) |
| $q_V, q_L, q$ | V and L mass flowrates (kg/s); HFM mass flux (kg/(s.m$^2$)) |
| $q_{V_k}, q_{L_k}$ | V and L mass flowrates of "k" (kg/s) |
| $Q^{flux}$ | Heat flux (W/m$^2$) |
| $Q$ | Heat rate (kJ/h) |
| $R$ | Ideal gas constant (8.314 J/(mol.K)) |
| $R_k$ | Chemical reaction rate in terms of "k" (mol/(m$^3$.s)) |
| $R_{k,mL}$ | Chemical reaction rate of "k" inside the liquid-filled part of HFM pores (mol/(m$^3$.s)) |
| $r$ | GLMC radial position (m) |
| $r_i, r_o$ | Internal/external HFM radius (m) |
| $r_{shell}$ | Outer radius of virtual cylindrical domain around each HFM (effective radius of shell-side) (m) |
| $r_i^+$ | Gas-membrane interface at the pore-side (m) |
| $Sa = N_{HF}\Pi d_o$ | Interfacial area per length (m) |
| $S_V, S_L$ | Sections of V and L flows (m$^2$) |
| $T^{amb}$ | Outside temperature (K) |
| $T, T_V, T_L$ | Temperature, V and L temperatures (K) |
| $T_{L,m}$ | Temperature at liquid–membrane interface (K) |
| $T_i$ | HFM inside temperature (K) |
| $T_m$ | HFM temperature (K) |
| $T_o$ | Shell-side temperature (K) |
| $U$ | Overall heat transfer coefficient (kJ/(h.m$^2$.K)) or (W/(m$^2$.K)) |
| $V_k^{flux}$ | V molar flow (flux) of "k" (mol/s) |
| $V_k$ | V molar flowrate of "k" (mol/s) |
| $v_i, v_o$ | Interstitial velocity of the stream at internal/external surface of HFM (m/s) |
| $v_V, v_L$ | V and L flow velocities (m/s) |
| $\overline{v}_V, \overline{v}_L$ | V and L flow average velocities (m/s) |
| $v_{r,i}$ | Velocity in radial direction inside HFM (m/s) |
| $v_{r,o}$ | Velocity in radial direction in the shell-side (m/s) |
| $\overline{v}_{z,i}$ | Average velocity in axial direction inside HFM (m/s) |
| $v_{z,i}$ | Velocity in axial direction inside HFM (m/s) |
| $\overline{v}_{z,o}$ | Average velocity in axial direction in the shell-side (m/s) |
| $v_{z,o}$ | Velocity in axial direction in the shell-side (m/s) |
| $X_L$ | Partial liquid penetration |
| $Y_k, X_k$ | V and L mol fractions of "k" |
| $Z$ | Compressibility factor |
| $z$ | GLMC axial position (m) |
| *Greek Symbols* | |
| $\Gamma_{P_V}, \Gamma_{P_L}$ | Isothermal compressibility of V and L at RVLE (kg/(Pa.m$^3$)) |
| $\Gamma_{T_V}, \Gamma_{T_L}$ | Isobaric expansivity of V and L at RVLE (kg/(K.m$^3$)) |
| $\Gamma_{V_k}, \Gamma_{L_k}$ | Species "k" mol expansivity of V and L at RVLE ((kg.s)/(mol.m$^3$)) |
| $\varepsilon$ | Membrane porosity |
| $\varepsilon_V$ | Fraction of total area available for gas flow |
| $H_k$ | Henry's law constant of "k" ((kPa.m$^3$)/mol) |
| $\theta$ | GLMC inclination (rad) |
| $\lambda_V, \lambda_L$ | V and L thermal conductivities (W/(m.K)) |
| $\mu_i$ | Dynamic viscosity of the predominant phase inside HFM (Pa.s) |
| $\mu_o$ | Dynamic viscosity of the predominant phase in shell-side (Pa.s) |
| $\varphi$ | Fiber volume fraction (packing ratio) (-) |
| $\Pi_k$ | Trans-membrane mass transfer coefficient of "k" (mol/(s.bar.m$^2$)) |
| $\rho_V, \rho_L$ | V and L densities (kg/m$^3$) |

| | |
|---|---|
| $\rho_i$ | *Density inside HFM (kg/m$^3$)* |
| $\rho_o$ | *Density in the shell-side (kg/m$^3$)* |
| $\tau$ | *Membrane tortuosity* |
| $\psi_V, \psi_L$ | *Shear stress on contact surfaces in V and L flows (Pa)* |
| $\Omega_E, \Omega_I$ | *External/internal heat transfer coefficients (kJ/(h.m$^2$.K)) or (W/(m$^2$.K))* |
| $\Omega_V, \Omega_L$ | *V and L heat transfer coefficient (kJ/(h.m$^2$.K)) or (W/(m$^2$.K))* |
| $\Omega_m$ | *HFM heat transfer coefficient (kJ/(h.m$^2$.K)) or (W/(m$^2$.K))* |
| $\Omega_i$ | *Heat transfer coefficient inside HFM (W/(m$^2$.K))* |
| $\Omega_o$ | *Heat transfer coefficient in the shell-side (W/(m$^2$.K))* |
| *Superscripts* | |
| *out* | *at outlet* |
| *in* | *at inlet* |

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
