# Peer review of "Carbon Capture from CO2-Rich Natural Gas via Gas-Liquid Membrane Contactors with Aqueous-Amine Solvents: A Review"

_2673-5628, doi:10.3390/gases2030007_

Round 1

Reviewer 1 Report

This paper extensively and critically reviewed the studies on the gas-liquid membrane contactor from principle, experiments and models, etc. in scientific and understandable language. In general, this is a high-quality review. However, the authors did not clearly point out the adaptability and shortcomings of different models when introducing various models and their corresponding formulas. It is expected authors can improve the paper from this aspect.

Author Response

This paper extensively and critically reviewed the studies on the gas-liquid membrane contactor from principle, experiments and models, etc. in scientific and understandable language. In general, this is a high-quality review. However, the authors did not clearly point out the adaptability and shortcomings of different models when introducing various models and their corresponding formulas. It is expected authors can improve the paper from this aspect.

Reviewer#1 asks to point out the adaptability and shortcomings of models. The authors clarify that, when the models were introduced in Sec. 4.1 and Sec. 4.2, adaptability and shortcomings of different models were clearly highlighted. Their corresponding formulas were presented in Sec. 4.3: Table 6, Table 7 and Table 8. Since models discussion were already presented in Sec. 4.1 and in Sec. 4.2, the aim of Table 6 is just to present 1D model equations, without any judgement. In the same way, the aim of Table 7 is just to present 2D model equations, while Table 8 presents 1D/2D model equations. However, it seems that Reviewer#1 did not link the content of Sec. 4.3 with the adaptability and shortcomings discussion of different models, already presented in Sec. 4.1 and in Sec. 4.2., what probably lead him to write this comment based on just the content of Sec. 4.3 and its respective tables.

In order to find a compromise solution and to comply with the Reviewer#1 request, we decided to change the text and do not change any table. In model discussions of Sec. 4.1 and of Sec. 4.2, we added a mention of which equations were adopted in each model. This way, we hope to comply with Reviewer#1 request.

There were several interventions in the text. Therefore, we ask Reviewer#1 to verify that in the Manuscript in order to avoid repeating them in the present letter, which would lead to a too big letter.

This query generated several changes in Sec. 4.1 and in Sec. 4.2.

Reviewer 2 Report

The overall paper titled “Carbon Capture from CO2-Rich Natural Gas via Gas-Liquid Membrane Contactors with Aqueous-Amine Solvents: A Review” is well written and comprehensively reviewed. The authors critically highlighted the literature about gas-liquid membrane contactors. Therefore, the paper can be accepted after following minor modifications.

1. The authors need to change the %mol to mol% where applicable.

2. Similarly, in some other places, authors used %w/w or different terms; it is suggested to change w/w %.

3. In Table 6, the authors should highlight the limitation. They should insert another column for limitaitons.

4. The authors need to check the grammar and language of the manuscript.

5. Format the nomenclature section.
